



# Palaeobiological evidence for Southern Hemisphere Younger Dryas and volcanogenic cold periods

Richard N Holdaway*[1,2,3]

[1]Palaecol Research Ltd, P.O. Box 16569, Hornby, Christchurch 8042, New Zealand.

[2]School of Biological Sciences, University of Canterbury, Private Bag 4800, Christchurch 8041, New Zealand.

[3]School of Earth and Environment, University of Canterbury, Private Bag 4800, Christchurch 8041, New Zealand.

*Correspondence to: Richard N. Holdaway (turnagra@gmail.com)



**Abstract.** Current consensus places a Southern Hemisphere post-glacial cooling episode earlier than the
Younger Dryas in the Northern Hemisphere. New Zealand sequences of glacial moraines and speleothem
isotopic data are generally interpreted as supporting the absence of a Southern Hemisphere Younger Dryas.
Radiocarbon age series of habitat specialist moa (Aves: Dinornithiformes) show, however, that a sudden return
to glacial climate in central New Zealand contemporary with the Younger Dryas. The cooling followed
significant warming, not cooling, during the period of the Antarctic Cold Reversal. In addition, the moa
sequence chronology also shows that the Oruanui (New Zealand) and Mt Takahe (Antarctica) volcanic eruptions
were contemporary with abrupt cooling events in New Zealand. The independent high spatial and temporal
resolution climate chronology reported here is contrary to an interhemispheric post-glacial climate see-saw
model.

## 1 Introduction

The present model for the interhemispheric relationships between climatic events during the last deglaciation
invokes a seesaw alternation of warming with returns to glacial conditions (Broecker, 1998; Barker et al., 2009).
The sequence of alternate warming and cooling in each hemisphere model is based predominantly on proxy
chronologies from ice and ocean cores and cave speleothems. From these chronologies, post-glacial warming
began c.18 ka BP, possibly triggered by a series of eruptions of Mt Takahe (Antarctica) (McConnell et al.,
2017), that was reversed temporarily first in the Southern Hemisphere by a cool episode (the Antarctic Cold
Reversal, ACR) which, as presently understood, preceded the abrupt return to glacial conditions of the European
Younger Dryas between 12.8 ka and 11.5 ka BP.

Whether there was a Southern Hemisphere cold period synchronous with the Younger Dryas has been
debated for many years, with discussions based partly on data from New Zealand's South Island. The island,
athwart the present dominant Southern Hemisphere temperate latitude westerly wind belt, is a key site for
Quaternary paleoclimate research. The position and strength of the westerlies has been interpreted from
speleothem stable isotopic records and glacial advances and retreats in the axial mountains. These data have
been used to both support (Denton & Hendy, 1994; Ivy-Ochs et al., 1999) and reject (Shulmeister et al., 2005;
Hajdas et al., 2006; Kaplan et al., 2010; Koffman et al., 2017; Putnam et al., 2010) a New Zealand Younger
Dryas episode. The moraines, speleothems, and pollen as climate proxies provide overviews of atmospheric and
conditions through time but they are either contentious, as with the origin of the Waihō Loop (Denton & Hendy,
1994; Mabin et al., 1996; Tovar et al., 2008), or, with speleothem isotopes and pollen, have been recovered from
relatively few sites, and are subject to different interpretations (Newnham, 1999; Singer et al., 1998). All,
especially the Kaipo Bog pollen record (Lowe & Hogg, 1986; Hajdas et al., 2006), depend on date-interpolated
stratigraphic chronologies. In addition, apart from rare instances such as the Kaipo Bog, the sites do not record
local environments in detail through space and time.

In contrast, the rich radiocarbon record for moa (Aves: Dinornithiformes), whose late Holocene
distributions indicate high levels of habitat specificity, provides a geographically extensive, independent,
intensive chronology of vegetation, west of the Main Divide (MD), in the western and northwestern South
Island, New Zealand over the past 25,000 years (Worthy, 1993a; Worthy, 1997; Worthy & Holdaway, 1993,
1994; Worthy & Roscoe, 2003). I used the published radiocarbon ages of four moa taxa characteristic of
alpine/glacial, lowland rain forest, and dry shrublands to develop a chronology of vegetation change through
time, comparable with interpretations of local speleothem $\delta^{18}$O palaeotemperature records and the European





Younger Dryas chronology from ice accumulation, temperature (Meese et al., 1994)) and methane (Brook et al.,
1996) records over the past 30 ka from the GISP2 ice core (Greenland).

**2 Materials and methods**
**2.1 Geographic settings**
**2.1.1 West Coast, South Island**
The West Coast fossil sites are within caves and cave systems in the 6000 ha of karst that abuts the coast south
of Charleston (Fig. 1). The area is bounded to the east by the 1500 m Paparoa Range. The limestone is c. 100 m
thick so there is little vertical development in the caverns, but horizontal passages can be up to 8 km long
(Worthy & Holdaway, 1993). The sites vary in present altitude from 10s to 300 m above sea level. The present
climate is mild, humid, and equable, with mean summer maxima of c. 18ºC and minima of c. 10 ºC (Worthy &
Holdaway, 1993). The dominant westerly winds bring precipitation from the Tasman Sea and beyond. Rainfall
varies from 2800 mm at the coast to 4000 mm at the base of the eastern ranges, with 8000 mm at the summit.
Annual sunshine hours range from 1700 in the south to nearly 2000 in the north. Fed by the ample moisture and
mild temperatures the Holocene vegetation has been lowland rain forest, with podocarp conifers emergent above
a canopy of broadleaf trees and an understory of shrubs and tree ferns.

**2.1.2 Honeycomb Hill**
The Honeycomb Hill Cave system of 13.7 km of passages, with 70 entrances, lies beneath 0.8 km$^2$ of temperate
rain forest (Worthy, 1993a) in the broad, shallow Oparara River basin in the north-western South Island (Fig. 1).
The basin is sheltered, and normally experiences only light winds, mostly from the west. Annual rainfall is
3000-4000 mm (Worthy, 1993a). Frosts are uncommon and temperatures are similar to those in the West Coast
study area. The present altitude of 300 m a.s.l. was c. 430 m during the glacial maximum.

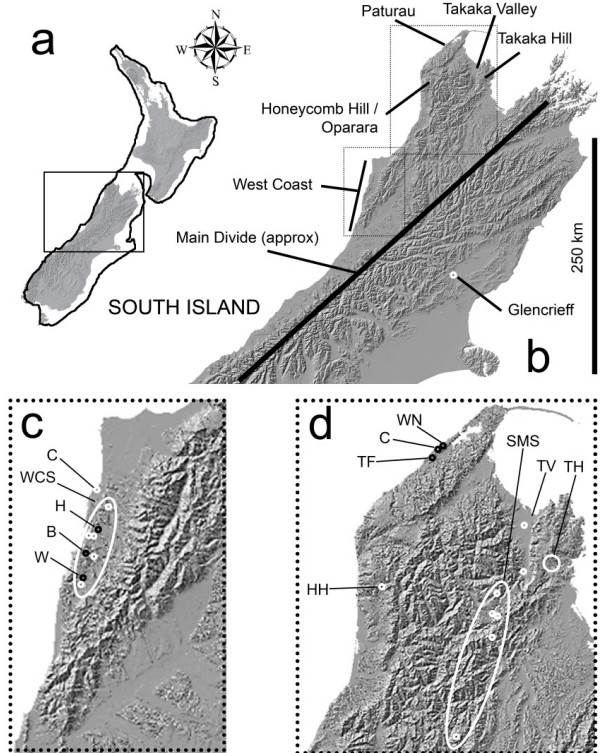


**Figure 1:** Distribution of fossil deposits and speleothems in the northern South Island, New Zealand. Fossil

deposits, black; speleothem sites, white. **A**, Location of study area in relation to present and Last Glacial

Maximum (solid line) coastlines. **B**, Western and northwestern study areas and Glencrieff site in relation to the

Main Divide. **C**, Detail of West Coast study area. C, Charleston;. WCS, West Coast fossil sites; H, Hollywood

Cave; B, Babylon Cave; W, Wasspretti Cave. **D**, Detail of Northwest Nelson study area. WN, Wet Neck Cave;

C, Creighton's Cave; TF, Twin Forks Cave; HH, Honeycomb Hill Cave system; SMS, Southern mountain sites;

TV, Takaka Valley; TH, Takaka Hill; A, Mt Arthur (Nettlebed Cave)

### 2.1.3 Takaka Hill

Takaka Hill (TH) is a roughly tear-drop-shaped (in plan) complex of ranges and marble plateaux which projects

40 km north from the high (1500-1800 m) mountainous core of northwest Nelson (Fig. 1). The southern end is a

narrow saddle reaching 1000 m, between the headwaters of the Waitui Stream to the west and the South Branch



of the Riwaka River to the east, which pinches it off from the southern mountains. The 'Hill' itself, c. 20 km
across at its widest point, is therefore in effect an island, bounded by steep escarpments to the east and west and
to the north (during the Holocene) by the sea. The western escarpment is the boundary with Takaka Valley (TV)
where there are fossil deposits in limestone caves: to the east the abrupt slopes fall to the Waimea Plains (Fig.
1), which have great depths of alluvium and no known Quaternary fossil deposits. Most of the fossil sites on TH
are presently at 800-900 m above sea level, but would have been up to 1000 m during the glaciation.

Although at present near sea level, TV would have been near 250 m at Last Glacial Maximum. During

the Weichselian-Otiran glaciation TH and TV were near the center of the larger glacial period New Zealand land
mass, much farther than at present from coastal and marine influences. To the north, extensive lowlands of the
land bridge occupied the western reaches of present Cook Strait. The land bridge allowed contact, at times,
between the faunas and floras of the North and South islands. The status of the Cook Strait land bridge as
alternatively a barrier to movement or a conduit has been much debated, e.g., (Worthy, 1993b; Worthy &
Holdaway, 1994).
Its roles as barrier or conduit probably changed with extrinsic events. For example, the Oruanui
eruption of Taupo Volcano 25.6 ka BP (Vandergoes et al., 2013) would have deposited thick ash on the bridge
(Vandergoes et al., 2013), destroying the vegetation (Oppenheimer, 2011), and making it uninhabitable by moa
or anything else. Under the cold, dry glacial climate revegetation would have taken millennia.
Much of the forest on TH was removed in the early 20th century. Surviving patches indicate that the
Holocene mixed forest was dominated by *Fuscospora* and *Lophozonia* southern beeches (both formerly in
*Nothofagus*) with emergent podocarps including *Podocarpus totara* and *P. laetus* (formerly *P. hallii* or *P.*
*cunninghamii*) (Worthy & Holdaway, 1994). The Holocene TV rain forest was much floristically richer than
that on TH. Rainfall in TV and on TH is high and TH regularly receives (southern) winter snow.

### 2.1.4 Glencrieff

The Glencrieff site (Fig. 1) (Worthy & Holdaway, 1996) is a small bog formed over a spring in a grazed field on
a gravel river terrace c. 1.6 km west of the well-known Pyramid Valley lake bed deposit (Holdaway & Worthy,
1997; Holdaway et al., 2014; Allentoft et al., 2014). The moa remains lay within a 1.2 m peat layer beneath the
turf. The peat in turn overlays fluid blue clay over a gravel base at c. 2.5 m. Pollen recovered from the peat
suggested that the local vegetation during moa deposition was tussock grasses (mostly *Chionochloa* spp.) with
the small conifer *Phyllocladus alpinus* and *Coprosma* spp. shrubland. As both *Chionochloa* and *Coprosma* are
both anemophilous and produce large amounts of mobile pollen, they may be over-represented in the deposit.

**2.1.5 Merino Cave, Annandale**
Merino Cave is at the foot of the slope in a small doline at 560 m at the southern end of a limestone plateau in
North Canterbury which rises to Mt Cookson (897 m) (Fig. 1). To the west, the eastern axial mountain ranges
reach >1700 m. The moa remains were excavated, with those of other birds, from the sides and floor of a
channel cut through a sediment fill (Worthy & Holdaway, 1995).

**2.2 Systematics – nomenclature of *Pachyornis* and *Euryapteryx***
It is necessary for clarity to briefly summarize recent changes in moa nomenclature because older major reviews
such as (Worthy & Holdaway, 2002) use older names and even species definitions. In *Pachyornis*, the validity
of *P. australis* was accepted on its morphology by (Worthy, 1989) and confirmed genetically later (Bunce et al.,
2009). Further analyses of the distributions and relationships of taxa within South Island *Pachyornis* led
(Rawlence et al., 2012) to question the validity of some identifications based on morphology. From this, and
stable isotope measurements, (Holdaway & Rowe, 2020) suggested that only *P. australis* was found west of the
Main Divide. For the present analysis, *Pachyornis* individuals west of the MD are treated as *P. australis*.
However, the distinctions detailed by (Rawlence et al., 2012) are followed below completeness.

In the genus *Euryapteryx*, re-examination of a type specimen necessitated name changes. *E. curtus* was

known until 2005 as *E. geranoides*. It was then renamed *E. gravis* (Worthy, 2005). Four years later it was
incorporated in an expanded concept of *E. curtus* (Bunce et al., 2009). However, the systematics of the genus
appear to be still not settled as two sympatric genetic clades of different body sizes from the Holocene of the
northern North Island's northern peninsula have been referred to as different species (Huynen et al., 2010) or
subspecies (Huynen et al., 2014). They are unlikely to have been subspecies as the forms were sympatric and
contemporary. The *Euryapteryx* specimens are listed provisionally here as *E. curtus*.




### 2.3 Habitat specificity of moa

### 2.3.1 General

The key observation that allows some moa to be used as vegetation proxies is that those taxa have been found
only in association with specific vegetation types in deposits of Holocene age. The Holocene vegetation pattern
is well known from existing examples and many pollen records.

South Island moa were formerly characterized in terms of "faunas", principally with respect to a

distinction between Holocene age "eastern" and "western" faunas (Worthy & Holdaway, 1993, 1994, 1996,
2002). The western "wet forest" fauna was based on the presence of *Anomalopteryx* and smaller individuals of
*Dinornis robustus*. Mid-sized individuals of *D. robustus* in western deposits were then identified as a separate
species (*D. novaezealandiae*) characteristic of wet forests. The eastern fauna was based on the presence of
*Euryapteryx* and *Pachyornis*. These genera have seen as characteristic of dry shrubland/forest mosaics (Worthy,
2000; Holdaway & Worthy, 1997; Worthy & Holdaway, 1994, 1996). Another emeid moa, *Emeus crassus*, was
also characteristic of the eastern fauna was but it has never been identified in deposits west of the MD (Worthy
& Holdaway, 2002), so provides no vegetation information for that area.

As noted above, these distinctions and the habitat preferences of the moa taxa were based on the

Holocene vegetation, which is well known both from surviving vegetation and pollen records. Habitat
requirements recognised here for the key taxa follow.

### 2.3.2 *P. australis*

This species indicates the presence of shrublands growing under a cool to cold climate. It occupied alpine
shrublands and fellfield in the Holocene (Worthy, 1989; Rawlence & Cooper, 2013; Worthy & Holdaway, 1994;
Rawlence et al., 2012), and is hypothesized to have occupied similar habitat during glaciations. The youngest
radiocarbon dated individuals are from higher altitudes than *P. elephantopus* (Rawlence & Cooper, 2013;
Rawlence et al., 2012).





### 2.3.3 *Anomalopteryx didiformis*

This small species indicates the presence of warm lowland to lower montane rain forest (Worthy & Holdaway, 1993, 1994, 2002). Carbon stable isotope ratios suggest that it fed around canopy gaps within the rain forest (Worthy & Holdaway, 2002).

### 2.3.4 *Euryapteryx curtus*

This species is found in Holocene in areas of lowland to lower montane dry forest and shrublands (Worthy & Holdaway, 2002). It is here taken to indicate the presence of such vegetation and also of seral vegetation with similar structure and composition.

## 2.4 Moa radiocarbon ages

### 2.4.1 Sources

Radiocarbon ages for moa obtained from the literature are listed, with ancillary data, in Tables 1 & 2. Those from the South Island West Coast are from (Worthy & Holdaway, 1993, 1994; Bunce et al., 2009); from Honeycomb Hill from (Worthy, 1993a; Rawlence et al., 2012; Bunce et al., 2009); from Glencrieff from references (Worthy & Holdaway, 1996; Rawlence et al., 2011); from Takaka Valley and Takaka Hill from (Worthy & Holdaway, 1994; Worthy & Roscoe, 2003; Rawlence et al., 2012; Bunce et al., 2009). Those from Annandale are in (Worthy & Holdaway, 1995).

### 2.4.2 Radiocarbon age calibration

All conventional radiocarbon ages were calibrated using OxCal4.4 (Ramsey, 2009) referenced to the SHCal20 curve (Hogg et al., 2020). The dates are presented as means or as means ± 1σ. At the chronological scales involved, symbol size in the figures generally encompasses the probability distribution for calibrated dates.





**2.5 Bayesian sequence analyses**
Sequences of moa radiocarbon ages and the timing of changes in representation were assessed using the
R_Sequence option in OxCal4.3 (Ramsey, 1995, 2009), again invoking the SHCal20 curve (Hogg et al., 2020).
Sequence starts and ends were plotted both as means and standard deviations and as probability distributions.
**2.6 Isotopes and other proxies**
Palaeoclimatic data were downloaded from the National Climatic Data Center of the National Oceanographic
and Atmospheric Administration (NOAA) www.ncdc.noaa.gov. The speleothem $\delta^{18}O$ data are from  (Hellstrom
et al., 1998), (Williams et al., 2005) and (Williams et al., 2010). Data for GISP2 ice accumulation are from
(Meese et al., 1994) and for GISP2 methane from (Brook et al., 1996).

Isotope, methane, and ice accumulation data were smoothed using local regression (LOESS), via the

LOESS option in PAST® (Hammer et al., 2001). A range of smoothing factors (Fig. 2-4) was tested to establish
which showed the best compromise between loss of pattern and excessive noise: a common smoothing factor of
0.03 was adopted. At all SF values, the fall in temperature signaled by the 8-speleothem $\delta^{18}O$ record coincided
with the start of the Younger Dryas in Greenland (Fig. 4).

**2.7 Eruption dates**
The date of the Oruanui (Taupo) super eruption is from radiocarbon ages in (Vandergoes et al., 2013)
recalibrated using the SHCal20 curve. The Mt Takahe eruption series date, based on Antarctic ice core
chronologies, is from (McConnell et al., 2017). The Mt Takahe eruptions coincided with a peak of warming
shown in the New Zealand $\delta^{18}O$ record, immediately preceding a return to low temperatures, rather than being at
the start of local warming (Fig. 2, 3).







**Table 1:** Conventional and calibrated (SHCal20 curve) radiocarbon ages (Before Present) for moa from West Coast,
Honeycomb Hill, Glencrieff, and Merino Cave deposits, South Island, New Zealand. *, Site 16 is also known as
Equinox Cave; NP, not published; ANDI, *Anomalopteryx didiformis*; PAEL, *Pachyornis elephantopus*; PAAU, *P.*
*australis*; *PAAU, P.* australis identified as *P. elephantopus* on morphology; EUCU, *Euryapteryx curtus*; EMCR,
*Emeus crassus*. Ages in italics, ANU laboratory series with different pretreatments (Rawlence et al., 2011).
Calibrated dates in square parentheses, failed $\chi^2$ test for combination. Date laboratory numbers in **bold** used in
sequence analysis. N/A, not available. References: B, Bunce *et al.* (2009); R11, Rawlence *et al.* (2011); R12,
Rawlence *et al*. (2012); W93, Worthy (1993a); W&H 93, Worthy & Holdaway (1993); W&H 95, Worthy &
Holdaway (1995); W&H 96, Worthy & Holdaway (1996).





| Taxon | Museum | Site | Lab no. | CRA | SD | Cal mean | SD | Median | Reference |
|---|---|---|---|---|---|---|---|---|---|
| **WEST COAST** | | | | | | | | | |
| ANDI | NMNZ S28055-61 | Madonna Cave Sites 1-8 | NZA2443 | 2197 | 86 | 2193 | 114 | 2141 | W & H 93 |
| PAEL | NMNZ S28064 | Madonna Cave Site 8 | NZA2505 | 14740 | 110 | 17985 | 164 | 18008 | W & H 93 |
| PAEL | NMNZ S28086 | Madonna Cave Site 14 | NZA2446 | 20680 | 160 | 24846 | 234 | 24861 | W & H 93 |
| PAAU | NMNZ S28192 | Te Ana Titi Cave | NZA2320 | 25070 | 260 | 29360 | 308 | 29345 | W & H 93 |
| EUCU | NMNZ S28083 | Madonna Cave Site 13 | NZA2779 | 11090 | 100 | 12971 | 101 | 12977 | W & H 93 |
| EUCU | NMNZ S28121 | Madonna Cave Site 16* | NZA2445 | 23780 | 210 | 27964 | 249 | 27924 | W & H 93 |
| PAAU | CMAv29445 | Charleston | OxA12431 | 14045 | 65 | 17054 | 122 | 17043 | B |
| **HONEYCOMB HILL CAVE SYSTEM** | | | | | | | | | |
| PAAU | *In situ* | Gradungula Passage | OxA12435 | 18925 | 80 | 22786 | 126 | 22800 | B |
| PAAU | *In situ*, NRS348 | Gradungula Passage | OxA20284 | 19575 | 80 | 23526 | 144 | 23517 | R 12 |
| PAAU | *In situ*, NRS350 | Gradungula Passage | OxA20285 | 20760 | 90 | 24979 | 141 | 24998 | R 12 |
| PAAU | NMNZ S25863.2 | Moa Cave Extension | OxA20366 | 17645 | 60 | 21254 | 131 | 21257 | R 12 |
| PAAU | NMNZ S25863.1 | Moa Cave Extension | OxA20367 | 19335 | 70 | 23262 | 190 | 23210 | R 12 |
| PAAU | NMNZ S25655 | Wren Wrecker | OxA20286 | 16860 | 75 | 20336 | 99 | 20344 | R 12 |
| PAAU | NMNZ S25868 | Cemetery | ANU1611 | 14730 | 170 | 17922 | 238 | 17953 | R 12 |
| PAAU | NMNZ S25867 | Cemetery | ANU1612 | 14950 | 150 | 18240 | 218 | 18220 | R 12 |
| PAAU | NMNZ S25864 | Cemetery | NZ7646 | 15000 | 200 | 18292 | 257 | 18283 | W 93 |
| *PAAU* | *N/A* | *Graveyard* | NZ6586 | 14029 | 138 | 16995 | 212 | 17005 | W 93 |
| *PAAU* | *N/A* | *Graveyard* | NZ6453 | 15677 | 163 | 18953 | 180 | 18940 | W 93 |
| *PAAU* | *N/A* | *Graveyard* | NZ7323 | 18600 | 230 | 22531 | 243 | 22520 | W 93 |
| *PAAU* | *N/A* | *Graveyard* | NZ7292 | 20600 | 450 | 24773 | 528 | 24767 | W 93 |
| *PAAU* | *N/A* | *Moa Cave* | NZ7642 | 13850 | 140 | 16746 | 211 | 16752 | W 93 |
| *PAAU* | *N/A* | *Moa Cave* | NZ6480 | 14194 | 140 | 17234 | 203 | 17222 | W 93 |
| *PAAU* | *N/A* | *Moa Cave* | NZA574 | 18300 | 170 | 22185 | 186 | 22199 | W 93 |
| *PAAU* | *N/A* | *Moa Cave* | NZ7647 | 18650 | 250 | 22577 | 256 | 22572 | W 93 |
| *PAAU* | *N/A* | *Moa Cave Extension* | NZ6589 | 14062 | 138 | 17047 | 209 | 17055 | W 93 |
| *PAAU* | *N/A* | *Cemetery* | NZ7675 | 12950 | 450 | 15391 | 722 | 15403 | W 93 |
| **GLENCRIEFF** | | | | | | | | | |
| PAEL | NP | Glencrieff "Peg 1" | **NZA4162** | 11898 | 82 | 13722 | 115 | 13709 | W & H 96 |
| PAEL | NMNZ 32670.9 | Glencrieff Square A1 | ANU1607 | 11230 | 210 | 13106 | 193 | 13106 | R 11 |
| | NMNZ 32670.9 | Glencrieff Square A1 | ANU4923 | 10750 | 80 | 12682 | 69 | 12698 | R 11 |
| | *NMNZ 32670.9* | *Glencrieff Square A1* | **ANU7612** | 10510 | 80 | 12377 | 177 | 12384 | R 11 |
| | Combined | | | | | 12645 | 64 | 12658 | |
| PAEL | NMNZ S32670.8 | Glencrieff Square A2 | ANU4937 | 9070 | 80 | 10161 | 135 | 10192 | R 11 |
| | *NMNZ S32670.8* | *Glencrieff Square A2* | **ANU7265** | 10680 | 70 | 12637 | 74 | 12654 | R 11 |
| | Combined | | | | | [11561] | 144 | 11563 | |
| PAEL | NMNZ S32670.3 | Glencrieff Square A1 | ANU1606 | 11490 | 80 | 13337 | 81 | 13337 | R 11 |
| | NMNZ S32670.3 | Glencrieff Square A1 | ANU4925 | 11180 | 70 | 13063 | 78 | 13077 | R 11 |
| | *NMNZ S32670.3* | *Glencrieff Square A1* | **ANU7614** | 10980 | 70 | 12880 | 83 | 12869 | R 11 |
| | Combined | | | | | [13109] | 45 | 13111 | |
| PAEL | *NMNZ S32670.7* | *Glencrieff Square B2/B3* | **ANU7610** | 11390 | 130 | 13260 | 116 | 13256 | R 11 |
| PAEL | NMNZ S32670.2 | Glencrieff Square B2 | ANU1605 | 10580 | 90 | 12487 | 169 | 12529 | R 11 |
| | NMNZ S32670.2 | Glencrieff Square B2 | ANU4924 | 10760 | 70 | 12694 | 57 | 12707 | R 11 |
| | *NMNZ S32670.2* | *Glencrieff Square B2* | **ANU7613** | 10610 | 80 | 12554 | 130 | 12583 | R 11 |
| | Combined | | | | | 12640 | 61 | 12654 | |
| EMCR | NMNZ S32690 | Glencrieff – Square A1 | **NZA4079** | 10470 | 130 | 12289 | 225 | 12296 | W & H 96; R 11 |
| EMCR | NMNZ S32688 | Glencrieff – Square A1 | **NZA4018** | 10480 | 120 | 12307 | 213 | 12312 | W & H 96; R 11 |
| **ANNANDALE** | | | | | | | | | |
| PAEL | NMNZ33402-3pt | Merino Cave - West | NZA3814 | 14150 | 140 | 17174 | 208 | 17176 | W & H 95 |
| PAEL | NMNZ33402-3pt | Merino Cave - West | NZA3815 | 19580 | 230 | 23503 | 262 | 23500 | W & H 95 |
| PAEL | NMNZ33402-3pt | Merino Cave - East | NZA3816 | 38200 | 980 | 42422 | 630 | 42378 | W & H 95 |
| PAEL | NMNZ33402-3pt | Merino Cave - East | NZA3884 | 22690 | 400 | 26901 | 403 | 26918 | W & H 95 |
| PAEL | NMNZ33402-3pt | Merino Cave - East | NZA4197 | 37820 | 810 | 42144 | 455 | 42157 | W & H 95 |
| PAEL | NMNZ33402-3pt | Merino Cave - East | NZA4447 | 14010 | 110 | 16974 | 181 | 16982 | W & H 95 |





**Table 2:** Radiocarbon ages for moa from the Takaka area, north-western South Island, New Zealand. References: B,
Bunce et al. (2009); R, Rawlence *et al*. (2012); W & H 94, Worthy & Holdaway (1994); W & R 03, Worthy &
Roscoe (2003).

| Taxon | Museum | Site | Lab no. | CRA | SD | Cal mean | SD | Median | Reference |
|---|---|---|---|---|---|---|---|---|---|
| ANDI | NMNZ S38943 | Takaka Fossil Cave | OxA12728 | 11575 | 45 | 13406 | 52 | 13407 | B |
| ANDI | | Takaka Fossil Cave | NZA11614 | 11354 | 60 | 13223 | 55 | 13221 | W & R 03 |
| ANDI | | Kairuru Extension | NZA3288 | 8274 | 72 | 9217 | 114 | 9213 | W & H 94 |
| ANDI | | Hawkes Cave | NZA3258 | 6656 | 141 | 7500 | 127 | 7503 | W & H 94 |
| ANDI | | Kairuru Extension | NZA3289 | 4072 | 59 | 4544 | 117 | 4524 | W & H 94 |
| ANDI | | Takaka Fossil Cave | NZA13547 | 1576 | 60 | 1429 | 65 | 1425 | W & R 03 |
| ANDI | | Irvine's Tomo | NZA3048 | 670 | 59 | 604 | 39 | 604 | W & H 94 |
| EUCU | | Takaka Valley | NZA3050 | 14080 | 100 | 17096 | 154 | 17093 | W & H 94 |
| EUCU | | Takaka Valley | NZA3051 | 13889 | 95 | 16805 | 153 | 16813 | W & H 94 |
| EUCU | | Takaka Hill | NZA1567 | 13400 | 130 | 16083 | 196 | 16082 | W & H 94 |
| EUCU | S39016 | Takaka | OxA12670 | 12525 | 50 | 14698 | 188 | 14677 | B |
| EUCU | | Takaka Fossil Cave | NZA13267 | 12450 | 65 | 14563 | 206 | 14547 | W & R 03 |
| EUCU | | Takaka Fossil Cave | NZA13266 | 12361 | 65 | 14407 | 223 | 14349 | W & R 03) |
| PAELN | S32425 | Predator Cave | OxA20336 | 32230 | 380 | 36616 | 441 | 36579 | R |
| PAAU | S28422 | Hawkes Cave | NZA3237 | 29011 | 312 | 33392 | 484 | 33444 | W & H 94 |
| PAELN | DM417E | Takaka Hill | OxA20292 | 20330 | 90 | 24390 | 143 | 24386 | R |
| PAEL | S27797 | Kairuru | NZA1568 | 18950 | 230 | 22836 | 271 | 22815 | W & H 94 |
| PAAU | NM unreg | Takaka Hill | OxA20290 | 18235 | 80 | 22160 | 92 | 22160 | R |
| PAELN | DM417E | Takaka Hill | OxA20293 | 14145 | 60 | 17184 | 93 | 17183 | R |
| PAEL | S28424 | Hawkes Cave | NZA3240 | 13470 | 94 | 16178 | 150 | 16179 | W & H 94 |
| PAAU | S33754 | Moa Trap Cave | OxA12669 | 10450 | 45 | 12303 | 136 | 12299 | B |
| PAAU | WO90.47 | Takaka Hill | OxA20291 | 10210 | 45 | 11814 | 89 | 11822 | R |
| PAAU | Av21331 | Bone Cave | OxA12430 | 10165 | 50 | 11716 | 134 | 11745 | B |
| PAAU | S27881 | Irvine's Tomo | NZA3049 | 28520 | 290 | 32687 | 470 | 32669 | W & H 94 |
| PAAU | S35298.1 | Commentary Cave | OxA20294 | 28050 | 300 | 32151 | 465 | 32062 | R |
| PAEL | NMNZ | Tarakohe | NZA3047 | 19520 | 130 | 23466 | 189 | 23472 | W & H 94 |


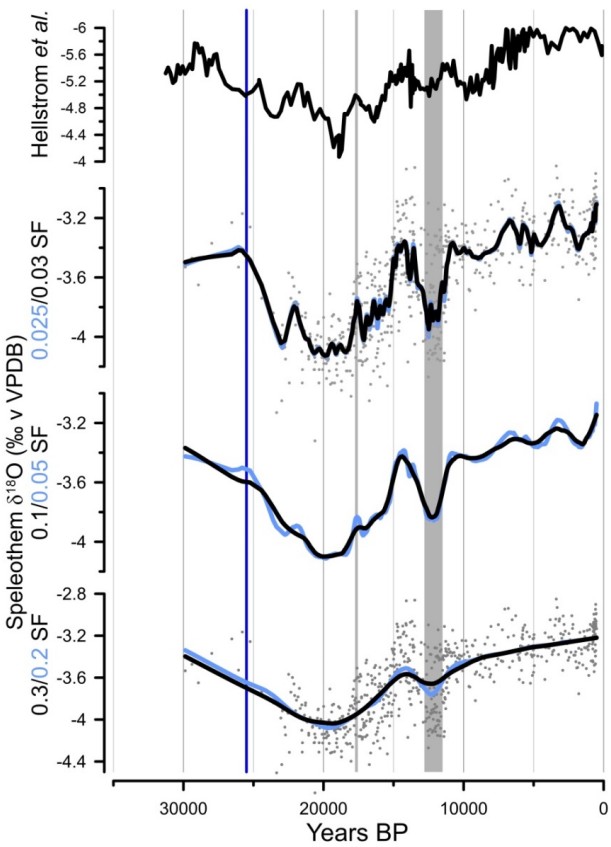

**Figure 2:** Effects of different smoothing factors on full sequence of speleothem $\delta^{18}$O records in western and

northwestern South Island, New Zealand. **A**, speleothem MD1, Mt Arthur. **B**, Integrated speleothem records from

eight caves near Paturau and on the West Coast, South Island, New Zealand; grey symbols, raw data; black, local

regression (LOESS) 0.03 smoothing factor (SF) (adopted for analyses); blue = 0.025 SF. **C**, As in B but black = 0.1

SF, blue = 0.05 SF; **D**, As in B, black = 0.3 SF, blue = 0.2 SF. Line at 25.6 ka BP, Oruanui (Taupo volcano, New

Zealand, eruption); at 17.7 ka BP, Mt Takahe, Antarctica, eruption series. Grey shading, European Younger Dryas.





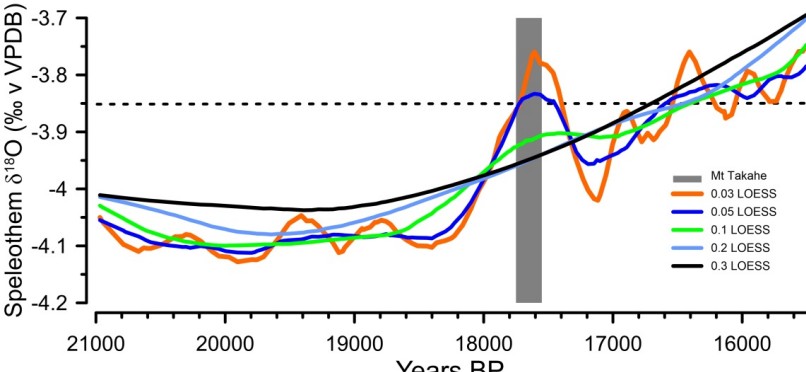


**Figure 3:** Effects of different smoothing factors (SF) on LOESS (local regression) of the 8-speleothem integrated

d18O record for the north-western South Island in the period encompassing the Mt Takahe (Antarctica) eruptiopns.

0.03 SF adopted for further analyses.


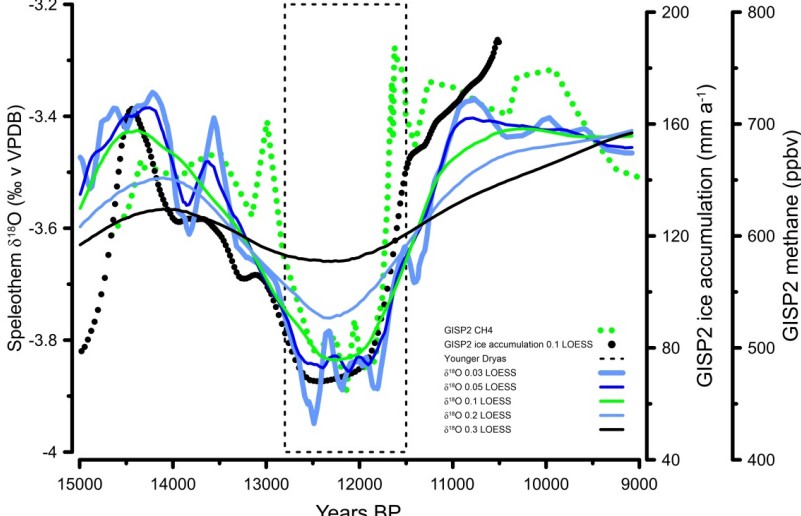


**Figure 4:** As in Fig. 3, but for the period encompassing the North Hemisphere Younger Dryas, in relation also to

GISP2 ice core methane and ice accumulation records (with 0.1 SF).






**3 Results**

Analysis of radiocarbon ages on bone gelatin of moa from cave sites at different altitudes in the north-western South Island, New Zealand, and a swamp site (Glencrieff) and cave site (Merino Cave, Annandale) east of the Main Divide showed changes in species representation in relation to each other, to the Oruanui super eruption (Taupo, New Zealand) and Mt Takahe eruptions (Antarctica), and to the period of the Younger Dryas (Fig. 5-8; Tables 1-3).

East of the MD, a population of *Pachyornis elephantopus* around Glencrieff was replaced by *Emeus crassus* at the onset of the Younger Dryas (Fig. 5, 6; Table 3), but both taxa were present in the area later in the Holocene (Holdaway et al. 2014). Only individuals of *Pachyornis* have been dated so far from Merino Cave. They were present around the time of the Oruanui eruption and after the Mt Takahe eruptions. West of the MD, apart from an individual > 30 ka old, deposition of *Pachyornis australis* in the Honeycomb Hill cave system began immediately after the Oruanui eruption (Fig. 5, 7; Table 4). Further south, in the West Coast area, two of the three available [14]C ages on *P. australis* fell immediately after the Oruanui and immediately before the Mt Takahe eruptions (Fig. 5A). The third, from a site at lower altitude, pre-dated the Oruanui eruption. One of the two available [14]C ages on *Euryapteryx curtus* from the West Coast pre-dated the Oruanui eruption; the other coincided with the start of the Younger Dryas (Fig. 5A). Despite significant numbers of *Anomalopteryx didiformis* in West Coast sites, only one, with a late Holocene age, has been [14]C-dated so far. Its presence was then assumed to indicate its presence in the Holocene only (Worthy & Holdaway, 1993).

Further north, [14]C ages for *P. australis*, *E. curtus*, and *A. didiformis* from Takaka Hill and the mountains to the south were clustered (Fig. 5B, C). Taking all the *Pachyornis* individuals as representing *P. australis* (Holdaway & Rowe, 2020), the species exhibited four pulses of presence on Takaka Hill (Fig. 5B). The first, at 38-34 ka BP, included the date for the single pre-Oruanui individual from Honeycomb Hill. As at Honeycomb Hill, there was a pulse of deposition after the Oruanui eruption, then none after 22 ka BP until after the Mt Takahe eruptions (Fig. 5B). *P. australis* was then replaced briefly by *A. didiformis* during the Antarctic Cold Reversal interval, only to return in the Younger Dryas period (Fig. 5, 8; Table 5). It vanished from Takaka Hill between 11.59 ka and 11.23 ka BP, synchronous with the end of the Younger Dryas (Fig. 5, 8; Table 5), after which it remained at higher altitudes in the mountains to the south, until its extinction (Fig. 5B). The *Pachyornis* individuals arrayed as presently identified (*P. australis*, *P. elephantopus* northern clade, *P. elephantopus* (*sensu lato*) appear sequentially in the Takaka area after both eruptions (Fig. 5C).





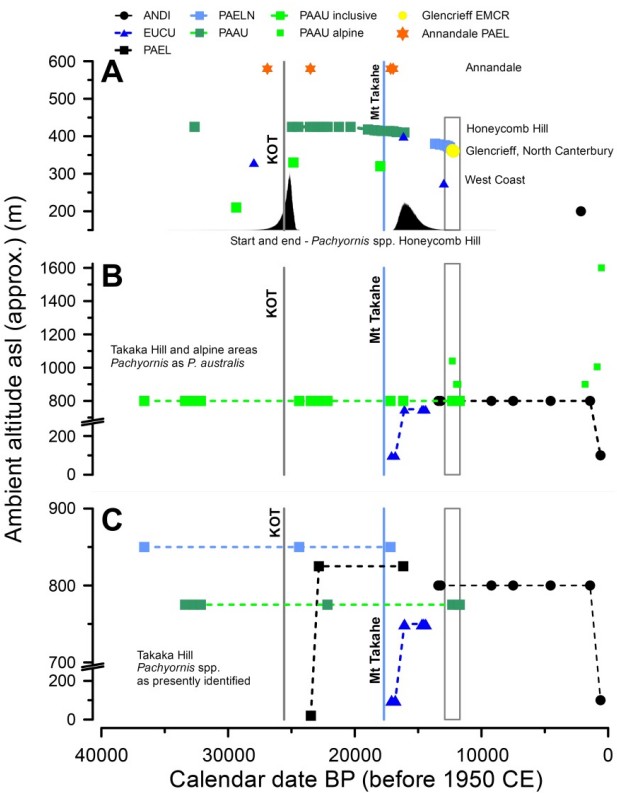


**Figure 5:** Species sequences of moa in the northern South Island, New Zealand over the past 40,000 years in
relation to the Oruanui and Mt Takahe eruptions and the Younger Dryas and to ambient altitude above sea level
(asl). Mean calibrated dates for moa in: **A**, the Honeycomb Hill cave system, the West Coast, and Glencrieff, with
Bayesian probabilities (shaded) of start and end of deposition of *Pachyornis australis* at Honeycomb Hill; B, the
Takaka area, all *Pachyornis* regarded as *P. australis*; and **C**, the Takaka area with *Pachyornis* as presently
identified. Abbreviations: KOT, Oruanui eruption; ANDI, *Anomalopteryx didiformis*; EUCU, *Euryapteryx curtus*;
PAEL, *Pachyornis elephantopus* sensu lato; PAELN, *P. elephantopus*, northern clade; PAAU, *P. australis* as
presently identified; PAAU inclusive, all *Pachyornis* west of the Main Divide regarded here as *P. australis*.







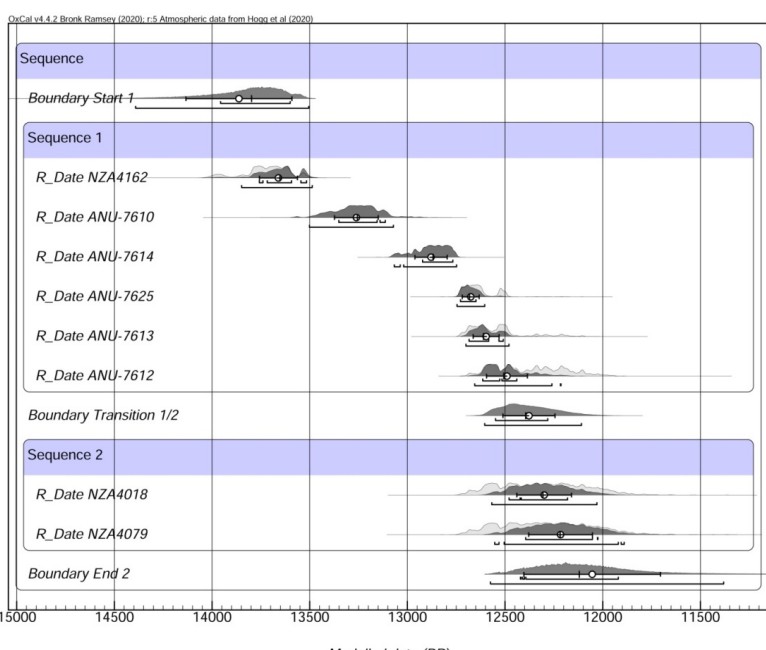

**Figure 6:** Bayesian modelled dates for transition between presence of *Pachyornis elephantopus* and *Emeus crassus*
in the Glencrieff deposit, eastern South Island, New Zealand.




Currently, there are no [14]C-dated *E. curtus* from the Takaka area from earlier than the Mt Takahe eruptions
(Fig. 5B). The short date series begins with individuals in the Takaka Valley, adjacent to the main part of the Cook
Strait land bridge (Fig. 8B), and only later on Takaka Hill, where its final date just precedes the brief presence of *A.*
*didiformis* itself just preceding the period of the European Younger Dryas.
Features in the moa date series and sequences aligned with the 8-speleothem $\delta^{18}$O paleotemperature
records and with both ice



| Name | Unmodelled (BP) | | | | | | | | | Modelled (BP) | | | | | | | | | Indices |
|---|---|---|---|---|---|---|---|---|---|---|---|---|---|---|---|---|---|---|---|
| | | | | | | | | | | | | | | | | | | | Amodel=122.6 |
| Show all | | | | | | | | | | | | | | | | | | | Aoverall=121.4 |
| Show structure | from | to | % | from | to | % | μ | σ | m | from | to | % | from | to | % | μ | σ | m | A |
| Curve SHCal20 | | | | | | | | | | | | | | | | | | | |
| Sequence | | | | | | | | | | | | | | | | | | | |
| Boundary Start 1 | *Start of Pachyornis* | | | | | | | | | 13956 | 13600 | 68.3 | 14391 | 13505 | 95.4 | 13862 | 271 | 13798 | |
| Sequence 1 | | | | | | | | | | | | | | | | | | | |
| R_Date NZA4162 | 13792 | 13606 | 68.3 | 14012 | 13508 | 95.4 | 13722 | 115 | 13709 | 13758 | 13516 | 68.3 | 13849 | 13487 | 95.4 | 13660 | 97 | 13653 | 98.9 |
| R_Date ANU-7610 | 13353 | 13111 | 68.3 | 13569 | 13020 | 95.4 | 13260 | 116 | 13256 | 13351 | 13113 | 68.3 | 13502 | 13072 | 95.4 | 13261 | 112 | 13256 | 100.8 |
| R_Date ANU-7614 | 12923 | 12767 | 68.3 | 13069 | 12748 | 95.4 | 12880 | 83 | 12869 | 12923 | 12767 | 68.3 | 13067 | 12748 | 95.4 | 12879 | 82 | 12869 | 100.2 |
| R_Date ANU-7625 | 12725 | 12619 | 68.3 | 12742 | 12487 | 95.4 | 12637 | 74 | 12654 | 12728 | 12649 | 68.3 | 12746 | 12604 | 95.4 | 12676 | 43 | 12682 | 109.1 |
| R_Date ANU-7613 | 12696 | 12485 | 68.3 | 12738 | 12177 | 95.4 | 12554 | 130 | 12583 | 12684 | 12509 | 68.3 | 12700 | 12480 | 95.4 | 12597 | 66 | 12611 | 115 |
| R_Date ANU-7612 | 12616 | 12103 | 68.3 | 12689 | 12023 | 95.4 | 12377 | 177 | 12384 | 12614 | 12440 | 68.3 | 12656 | 12214 | 95.4 | 12490 | 105 | 12507 | 117.5 |
| Boundary Transition 1/2 | *Pachyornis to Emeus* | | | | | | | | | 12549 | 12282 | 68.3 | 12604 | 12109 | 95.4 | 12378 | 133 | 12395 | |
| Sequence 2 | | | | | | | | | | | | | | | | | | | |
| R_Date NZA4018 | 12609 | 12093 | 68.3 | 12691 | 11934 | 95.4 | 12307 | 213 | 12312 | 12480 | 12180 | 68.3 | 12568 | 12030 | 95.4 | 12300 | 140 | 12306 | 111 |
| R_Date NZA4079 | 12606 | 12090 | 68.3 | 12694 | 11882 | 95.4 | 12289 | 225 | 12296 | 12395 | 12025 | 68.3 | 12553 | 11890 | 95.4 | 12216 | 163 | 12218 | 105.8 |
| Boundary End 2 | *End of dated sequence* | | | | | | | | | 12422 | 11920 | 68.3 | 12575 | 11381 | 95.4 | 12054 | 350 | 12120 | |

**Table 3:** Bayesian results for dates of moa species (*Pachyornis* by *Emeus*) replacement at Glencrieff, South Island, New Zealand.


accumulation and methane in the GISP2 ice core (Fig. 9). Approaching the Younger Dryas (as defined by the
GISP2 methane curve), the 8-speleothem $\delta^{18}$O record follows the same pattern as that of the GISP2 ice
accumulation, falling from a peak near 17 ka BP to a sudden drop at the onset of the Younger Dryas (Fig.





9A). Two other features of the 8-speleothem curve of note are a sudden drop at the time of the Oruanui
eruption (Fig. 9A), and a steep rise starting c. 18 ka BP, peaking at the Mt Takahe eruptions before falling
and not reaching the same value again until c. 15 ka BP (Fig. 9B, 10). Neither feature appears in the GISP2
ice accumulation curve.

Bayesian sequence analysis indicated that deposition of *P. australis* ceased at Honeycomb Hill two

centuries after the post-glacial rise in $\delta^{18}$O, at c. 3.85‰ (Fig. 9, 10). The species reappeared at c. 17.4 ka BP,
as $\delta^{18}$O fell below c. 3.85‰ (Fig. 10) and persisted until c. 16.5 ka BP. The gap in presence of *P. australis* in
the Honeycomb Hill record bracketed the Mt Takahe eruptions, ceasing as the climate warmed and returning
as it cooled immediately after the eruptions.

The smoothed local $\delta^{18}$O and GISP2 curves showed similar patterns before and after the Younger

Dryas (Fig. 9, 11). *Anomalopteryx didiformis* appeared briefly on Takaka Hill before the Younger Dryas,
indicating that rain forest had reached at least 800 m a.s.l. during the post-glacial warming. It was then
replaced on Takaka Hill during the European Younger Dryas period (Fig. 9, 11). *A. didiformis* reappeared in
the early Holocene (Fig. 9, 11).





| Name | Unmodelled (BP) from | to | % | from | to | % | μ | σ | m | Modelled (BP) from | to | % | from | to | % | μ | σ | m | Y | A |
|---|---|---|---|---|---|---|---|---|---|---|---|---|---|---|---|---|---|---|---|---|
| **Post KOT sequence** | | | | | | | | | | | | | | | | | | | | $A_{model}$=124.8 |
| Show all | | | | | | | | | | | | | | | | | | | | $A_{overall}$=122.3 |
| Show structure | | | | | | | | | | | | | | | | | | | Acomb | A |
| Curve SHCal20 | | | | | | | | | | | | | | | | | | | Y | |
| Sequence | | | | | | | | | | | | | | | | | | | | |
| Boundary Start 1 | | | | | | | | | | 25592 | 24870 | 68.3 | 26492 | 24684 | 95.4 | 25400 | 512 | 25271 | -4.5 | |
| Sequence 1 | *Arrival of P. australis* | | | | | | | | | | | | | | | | | | | |
| R_Date OxA20285 | 25156 | 24886 | 68.3 | 25220 | 24679 | 95.4 | 24979 | 141 | 24998 | 25139 | 24842 | 68.3 | 25202 | 24656 | 95.4 | 24952 | 146 | 24973 | -4.5 | 96.1 |
| R_Date NZ7292 | 25275 | 24185 | 68.3 | 25775 | 23815 | 95.4 | 24773 | 528 | 24767 | 24881 | 24186 | 68.3 | 25031 | 23831 | 95.4 | 24461 | 328 | 24482 | -4.5 | 108.4 |
| R_Date OxA20284 | 23760 | 23366 | 68.3 | 23793 | 23250 | 95.4 | 23522 | 150 | 23515 | 23763 | 23376 | 68.3 | 23795 | 23281 | 95.4 | 23538 | 146 | 23530 | -4.5 | 101.5 |
| R_Date OxA20367 | 23317 | 23053 | 68.3 | 23710 | 23003 | 95.4 | 23262 | 190 | 23210 | 23279 | 23065 | 68.3 | 23664 | 22993 | 95.4 | 23206 | 133 | 23186 | -4.5 | 108 |
| R_Date OxA12435 | 22956 | 22690 | 68.3 | 22991 | 22551 | 95.4 | 22786 | 126 | 22800 | 22974 | 22767 | 68.3 | 23000 | 22605 | 95.4 | 22833 | 109 | 22857 | -4.5 | 107.4 |
| R_Date NZ7647 | 22867 | 22359 | 68.3 | 23022 | 22066 | 95.4 | 22577 | 256 | 22572 | 22796 | 22466 | 68.3 | 22914 | 22347 | 95.4 | 22625 | 153 | 22625 | -4.5 | 116.3 |
| R_Date NZ7323 | 22810 | 22305 | 68.3 | 22973 | 22079 | 95.4 | 22531 | 243 | 22520 | 22560 | 22265 | 68.3 | 22750 | 22120 | 95.4 | 22423 | 153 | 22418 | -4.5 | 112.3 |
| R_Date NZA574 | 22372 | 22051 | 68.3 | 22528 | 21776 | 95.4 | 22185 | 186 | 22199 | 22330 | 22028 | 68.3 | 22453 | 21788 | 95.4 | 22141 | 171 | 22159 | -4.5 | 101.7 |
| R_Date OxA20366 | 21395 | 21141 | 68.3 | 21463 | 20984 | 95.4 | 21254 | 131 | 21257 | 21395 | 21142 | 68.3 | 21463 | 20984 | 95.4 | 21254 | 131 | 21258 | -4.5 | 100 |
| R_Date OxA20286 | 20451 | 20257 | 68.3 | 20516 | 20134 | 95.4 | 20336 | 99 | 20344 | 20451 | 20256 | 68.3 | 20516 | 20134 | 95.4 | 20336 | 99 | 20344 | -4.5 | 99.9 |
| R_Date NZ6453 | 19096 | 18784 | 68.3 | 19384 | 18655 | 95.4 | 18953 | 180 | 18940 | 19094 | 18787 | 68.3 | 19350 | 18667 | 95.4 | 18962 | 169 | 18944 | -4.5 | 101.4 |
| R_Date NZA7646 | 18639 | 18063 | 68.3 | 18756 | 17806 | 95.4 | 18292 | 257 | 18283 | 18654 | 18210 | 68.3 | 18754 | 18110 | 95.4 | 18442 | 175 | 18475 | -4.5 | 108.9 |
| R_Date ANU-1612 | 18599 | 18020 | 68.3 | 18647 | 17868 | 95.4 | 18240 | 218 | 18220 | 18296 | 18031 | 68.3 | 18591 | 17880 | 95.4 | 18191 | 160 | 18192 | -4.5 | 112 |
| R_Date ANU-1611 | 18222 | 17782 | 68.3 | 18268 | 17422 | 95.4 | 17922 | 238 | 17953 | 18166 | 17742 | 68.3 | 18213 | 17458 | 95.4 | 17870 | 207 | 17900 | -4.5 | 100.9 |
| R_Date NZ6480 | 17382 | 17047 | 68.3 | 17766 | 16861 | 95.4 | 17234 | 203 | 17222 | 17417 | 17133 | 68.3 | 17721 | 17032 | 95.4 | 17318 | 166 | 17302 | -4.5 | 102.2 |
| R_Date NZ6589 | 17315 | 16900 | 68.3 | 17400 | 16616 | 95.4 | 17047 | 209 | 17055 | 17253 | 16986 | 68.3 | 17365 | 16849 | 95.4 | 17107 | 132 | 17101 | -4.5 | 117.7 |
| R_Date NZ6586 | 17281 | 16818 | 68.3 | 17377 | 16585 | 95.4 | 16995 | 212 | 17005 | 17086 | 16808 | 68.3 | 17218 | 16635 | 95.4 | 16934 | 143 | 16947 | -4.5 | 112.6 |
| R_Date NZ7642 | 16988 | 16564 | 68.3 | 17114 | 16293 | 95.4 | 16746 | 211 | 16752 | 16901 | 16543 | 68.3 | 17015 | 16354 | 95.4 | 16695 | 173 | 16701 | -4.5 | 106 |
| R_Date NZ7675 | 16161 | 14551 | 68.3 | 16840 | 14029 | 95.4 | 15391 | 722 | 15403 | 16672 | 15764 | 68.3 | 16910 | 15100 | 95.4 | 16081 | 486 | 16158 | -4.5 | 82.4 |
| Boundary End 1 | *P. australis extirpation* | | | | | | | | | 16467 | 15200 | 68.3 | 16804 | 14158 | 95.4 | 15634 | 712 | 15755 | -4.5 | |

**Table 4:** Bayesian results for dates of moa species succession at Honeycomb Hill, Oparara, South Island, New Zealand.

327

328





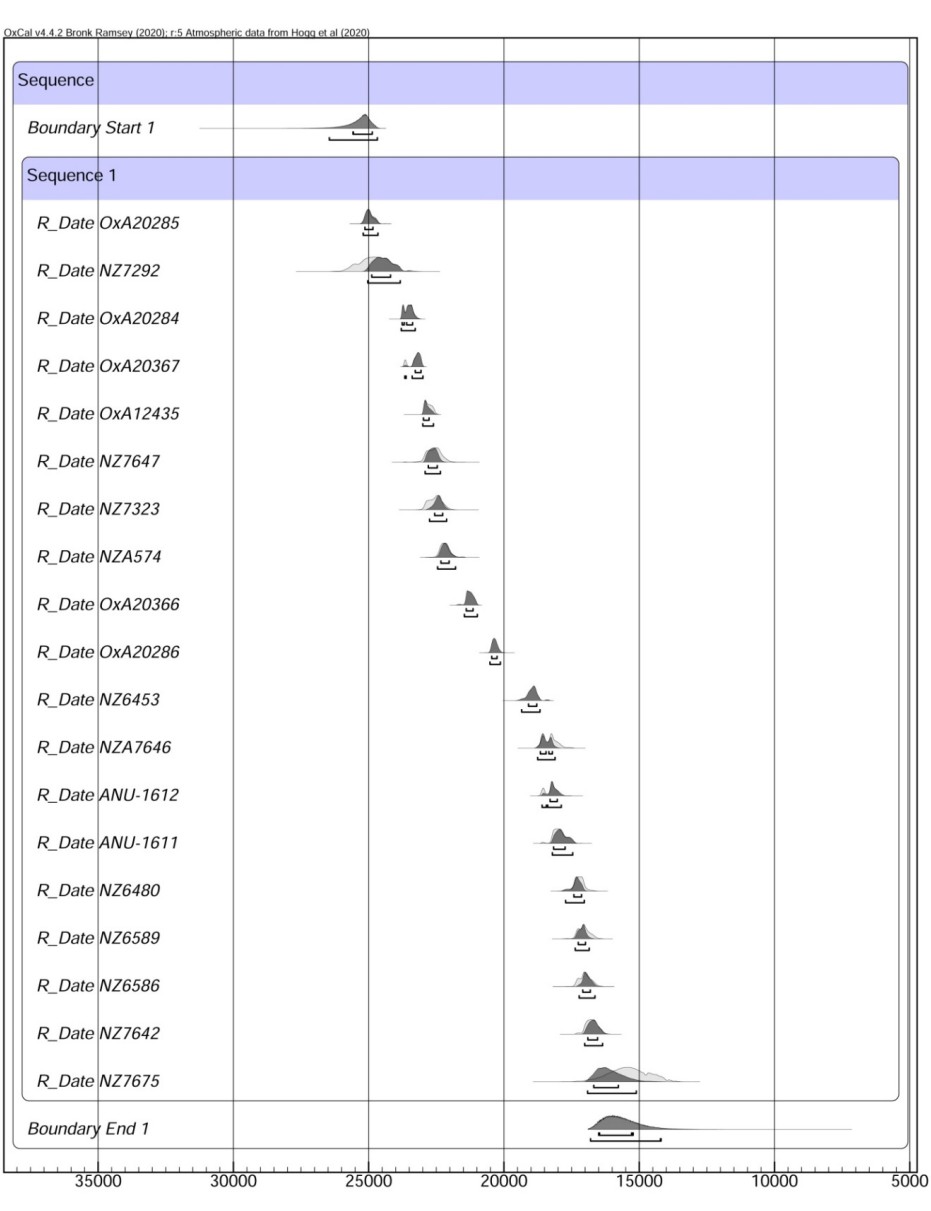

**Figure 7**: Bayesian modelled dates for the first and last *Pachyornis australis* after 30 ka BP in deposits in
the Honeycomb Hill Cave system, South Island, New Zealand.



| Name | Unmodelled (BP) | | | | | | | | | Modelled (BP) | | | | | | | | | Indices |
|---|---|---|---|---|---|---|---|---|---|---|---|---|---|---|---|---|---|---|---|
| | from | to | % | from | to | % | μ | σ | m | from | to | % | from | to | % | μ | σ | m | $A_{model}=103.2$ $A_{overall}=102.8$ A |
| Show all | | | | | | | | | | | | | | | | | | | |
| Show structure | | | 68.3 | | | 95.4 | | | | | | 68.3 | | | 95.4 | | | | |
| Curve SHCal20 | | | | | | | | | | | | | | | | | | | |
| Boundary Start 1 | First *Pachyornis* | | | | | | | | | 25654 | 24259 | 68.3 | 28209 | 24108 | 95.4 | 25468 | 1322 | 25039 | |
| Sequence 1 | | | | | | | | | | | | | | | | | | | |
| R_Date OxA20292 | 24532 | 24242 | 68.3 | 24672 | 24123 | 95.4 | 24390 | 143 | 24386 | 24505 | 24218 | 68.3 | 24657 | 24101 | 95.4 | 24366 | 143 | 24357 | 99.8 |
| R_Date NZA1568 | 23015 | 22530 | 68.3 | 23679 | 22367 | 95.4 | 22836 | 271 | 22815 | 23014 | 22529 | 68.3 | 23675 | 22368 | 95.4 | 22837 | 271 | 22815 | 100.1 |
| R_Date OxA20290 | 22254 | 22068 | 68.3 | 22345 | 21980 | 95.4 | 22160 | 92 | 22160 | 22254 | 22068 | 68.3 | 22345 | 21980 | 95.4 | 22160 | 92 | 22160 | 100.1 |
| R_Date OxA20293 | 17278 | 17069 | 68.3 | 17359 | 17025 | 95.4 | 17184 | 93 | 17183 | 17277 | 17069 | 68.3 | 17358 | 17025 | 95.4 | 17184 | 93 | 17182 | 99.9 |
| R_Date NZA3240 | 16320 | 16030 | 68.3 | 16489 | 15870 | 95.4 | 16178 | 150 | 16179 | 16380 | 16108 | 68.3 | 16547 | 15994 | 95.4 | 16259 | 139 | 16254 | 96.8 |
| Boundary Transition 1/2 | *P. australis* to *E. curtus* | | | | | | | | | 16236 | 15902 | 68.3 | 16400 | 15721 | 95.4 | 16063 | 169 | 16067 | |
| Sequence 2 | | | | | | | | | | | | | | | | | | | |
| R_Date NZA1567 | 16275 | 15884 | 68.3 | 16480 | 15699 | 95.4 | 16083 | 196 | 16082 | 16064 | 15750 | 68.3 | 16225 | 15598 | 95.4 | 15902 | 158 | 15901 | 86.1 |
| R_Date OxA12670 | 14958 | 14514 | 68.3 | 15015 | 14322 | 95.4 | 14698 | 188 | 14677 | 14980 | 14602 | 68.3 | 15049 | 14475 | 95.4 | 14791 | 158 | 14838 | 105.3 |
| R_Date NZA13267 | 14831 | 14319 | 68.3 | 14941 | 14201 | 95.4 | 14563 | 206 | 14547 | 14821 | 14325 | 68.3 | 14880 | 14255 | 95.4 | 14550 | 172 | 14529 | 107.9 |
| R_Date NZA13266 | 14791 | 14125 | 68.3 | 14837 | 14076 | 95.4 | 14407 | 223 | 14349 | 14383 | 14120 | 68.3 | 14781 | 14072 | 95.4 | 14312 | 155 | 14280 | 109.4 |
| Boundary Transition 2/3 | *E. curtus* to *A. didiformis* | | | | | | | | | 14118 | 13413 | 68.3 | 14374 | 13339 | 95.4 | 13836 | 303 | 13813 | |
| Sequence 3 | | | | | | | | | | | | | | | | | | | |
| R_Date OxA12728 | 13458 | 13352 | 68.3 | 13496 | 13313 | 95.4 | 13406 | 52 | 13407 | 13454 | 13347 | 68.3 | 13493 | 13312 | 95.4 | 13402 | 51 | 13401 | 100.1 |
| R_Date NZA11614 | 13296 | 13167 | 68.3 | 13316 | 13108 | 95.4 | 13223 | 55 | 13221 | 13298 | 13170 | 68.3 | 13320 | 13110 | 95.4 | 13227 | 54 | 13226 | 99.8 |
| Boundary Transition 3/4 | *A. didiformis* to *P. australis* | | | | | | | | | 13220 | 12443 | 68.3 | 13283 | 12196 | 95.4 | 12757 | 322 | 12775 | |
| Sequence 4 | | | | | | | | | | | | | | | | | | | |
| R_Date OxA12669 | 12477 | 12099 | 68.3 | 12594 | 12063 | 95.4 | 12303 | 136 | 12299 | 12472 | 12094 | 68.3 | 12585 | 12019 | 95.4 | 12276 | 135 | 12271 | 98.4 |
| R_Date OxA20291 | 11925 | 11735 | 68.3 | 11975 | 11525 | 95.4 | 11810 | 103 | 11820 | 11926 | 11780 | 68.3 | 11987 | 11651 | 95.4 | 11848 | 71 | 11850 | 106.4 |
| R_Date OxA12430 | 11872 | 11635 | 68.3 | 11930 | 11343 | 95.4 | 11716 | 134 | 11745 | 11839 | 11636 | 68.3 | 11883 | 11345 | 95.4 | 11706 | 119 | 11733 | 105 |
| Boundary Transition 4/5 | *P. australis* to *A. didiformis* | | | | | | | | | 11807 | 10803 | 68.3 | 11840 | 9562 | 95.4 | 11021 | 661 | 11233 | |
| Sequence 5 | | | | | | | | | | | | | | | | | | | |
| R_Date NZA3288 | 9398 | 9032 | 68.3 | 9421 | 9021 | 95.4 | 9217 | 114 | 9213 | 9397 | 9032 | 68.3 | 9420 | 9020 | 95.4 | 9216 | 114 | 9212 | 100 |
| R_Date NZA3258 | 7658 | 7340 | 68.3 | 7777 | 7180 | 95.4 | 7500 | 127 | 7503 | 7658 | 7357 | 68.3 | 7776 | 7182 | 95.4 | 7500 | 127 | 7503 | 99.9 |
| R_Date NZA3289 | 4780 | 4417 | 68.3 | 4818 | 4301 | 95.4 | 4544 | 117 | 4524 | 4781 | 4417 | 68.3 | 4818 | 4303 | 95.4 | 4544 | 117 | 4523 | 99.8 |
| R_Date NZA13547 | 1514 | 1359 | 68.3 | 1538 | 1310 | 95.4 | 1429 | 65 | 1425 | 1516 | 1365 | 68.3 | 1540 | 1309 | 95.4 | 1433 | 65 | 1431 | 99.4 |
| Boundary End 5 | | | | | | | | | | 1498 | -72 | 68.3 | 1559 | -3229 | 95.4 | 94 | 1604 | 644 | |

**Table 5** Bayesian results for dates of moa species succession in the Takaka area, South Island, New Zealand.



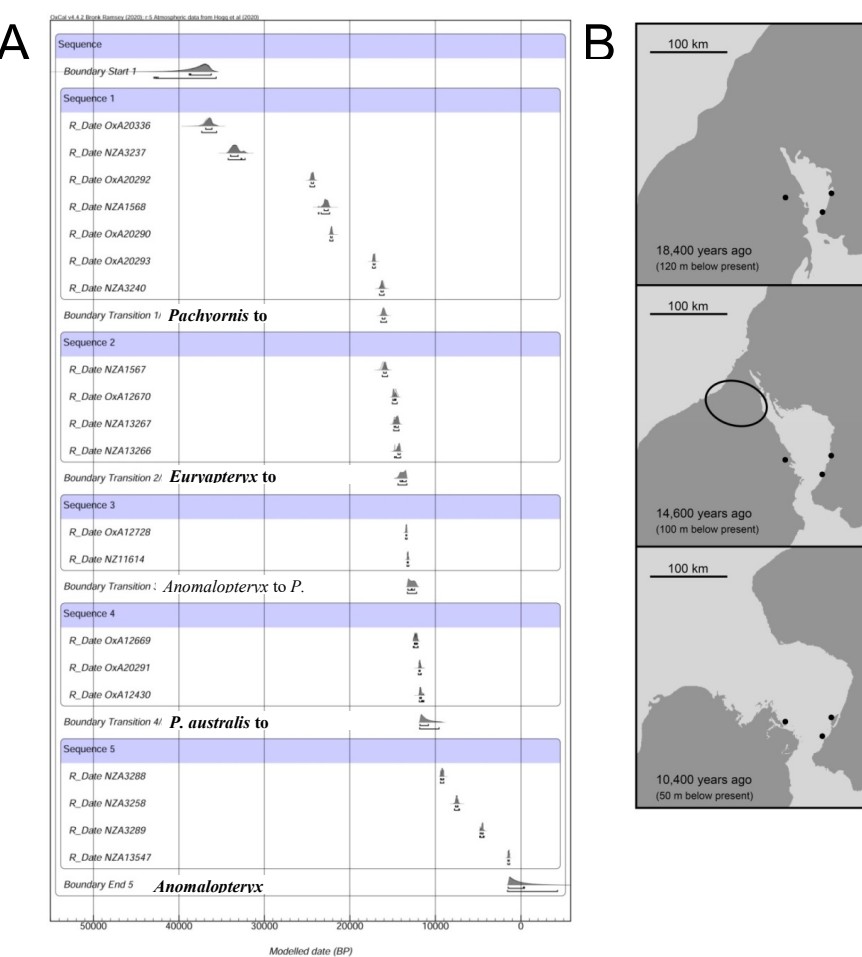

**Figure 8:** Bayesian modelled dates for transitions between moa taxa in the Takaka area, South Island, New Zealand and their geographic context. **A**, Bayesian modelled transition dates. **B**, Extent of the Cook Strait land bridge and when it was severed by rising post-glacial sea level. Bathymetric data from New Zealand Hydrographic Charts: NZ46, Cook Strait Narrows, 1:200000, published Jan 1989, Hydrographer RNZN. New Edition 2000, Land Information New Zealand; NZ48, Cook Strait Western Approaches, 1:400000, published Apr 1998, Hydrographer RNZN. New Edition Apr 2000, Land Information New Zealand.



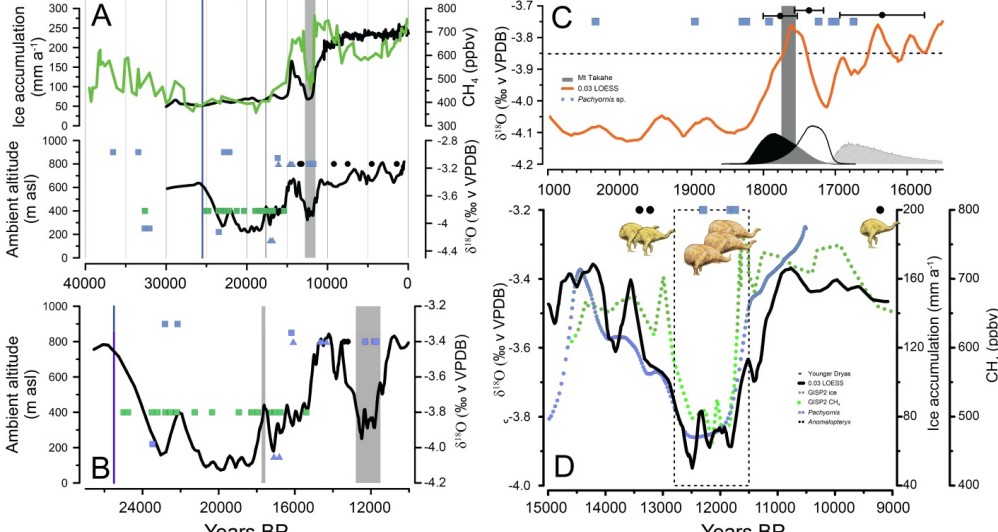

**Figure 9:** Moa taxa responded to the Younger Dryas climate shifts and volcanic events by tracking the resulting
geographic and altitudinal changes in their habitats. **A**, comparison of GISP2 (Greenland) ice accumulation (black)
and methane (green) (upper curves) with (lower curve) integrated speleothem $\delta^{18}O$ records for the northwestern
South Island and mean calibrated dates for moa at Honeycomb Hill and near Takaka. Vertical lines, L-R: Oruanui
eruption; Mt Takahe eruptions; Younger Dryas. **B**, detail of moa dates in comparison with northwestern South
Island speleothem $\delta^{18}O$. **C**, detail of dates for *Pachyornis australis* at Honeycomb Hill in relation to $\delta^{18}O$ record and
period of Mt Takahe (Antarctica) eruptions (shaded bar). Bayesian modelled dates for end of *P. australis* presence
before Mt Takahe, resumption of presence post-Mt Takahe, and extirpation of *P. australis* at Honeycomb Hill
indicated by means ± 1σ and posterior probability distributions. **D**, alternation of wet forest *Anomalopteryx*
*didiformis* and alpine/glacial vegetation *P. australis* on Takaka Hill 15-9 ka BP, in relation to local $\delta^{18}O$ record, and
GISP2 ice accumulation and methane. Black circles, *A. didiformis*; blue squares, *Pachyornis* spp.; green squares, P.
australis; blue triangles, *Euryapteryx curtus*.





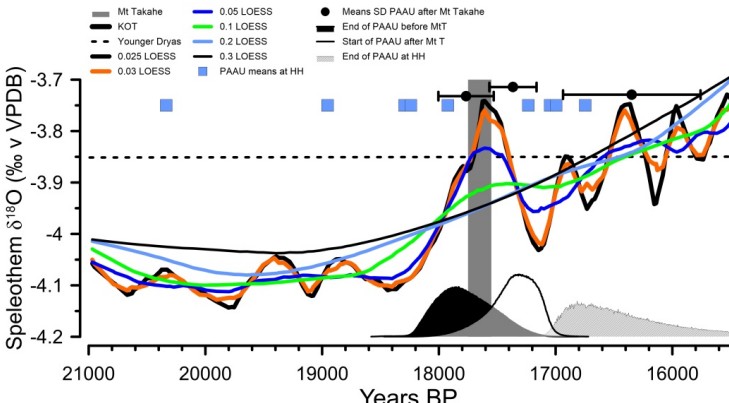

**Figure 10:** Detail of integrated speleothem oxygen isotope record around the Mt Takahe (Antarctica) eruptions (Figure 3, panel C) at various LOESS smoothing factors, with dates of *Pachyornis australis* individuals collected from the Honeycomb Hill cave system deposits.

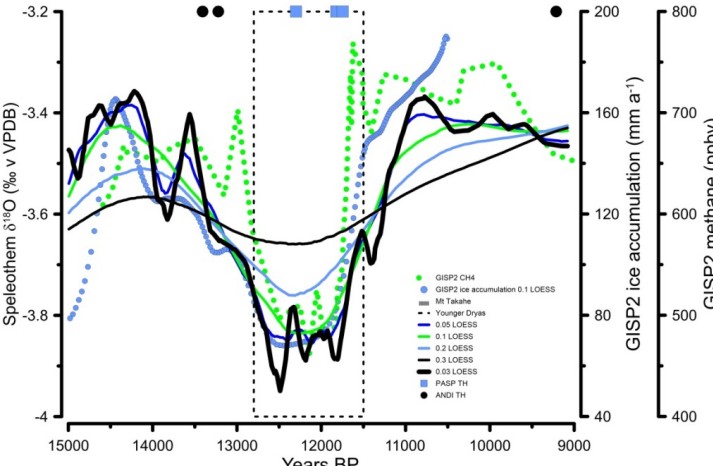

**Figure 11:** Detail of integrated speleothem oxygen isotope record around the Antarctic Cold Reversal and Younger Dryas periods (Figure 3, panel D) at various LOESS smoothing factors, with dates of *Anomalopteryx didiformis* and *Pachyornis australis* individuals collected from Takaka Hill cave deposits.



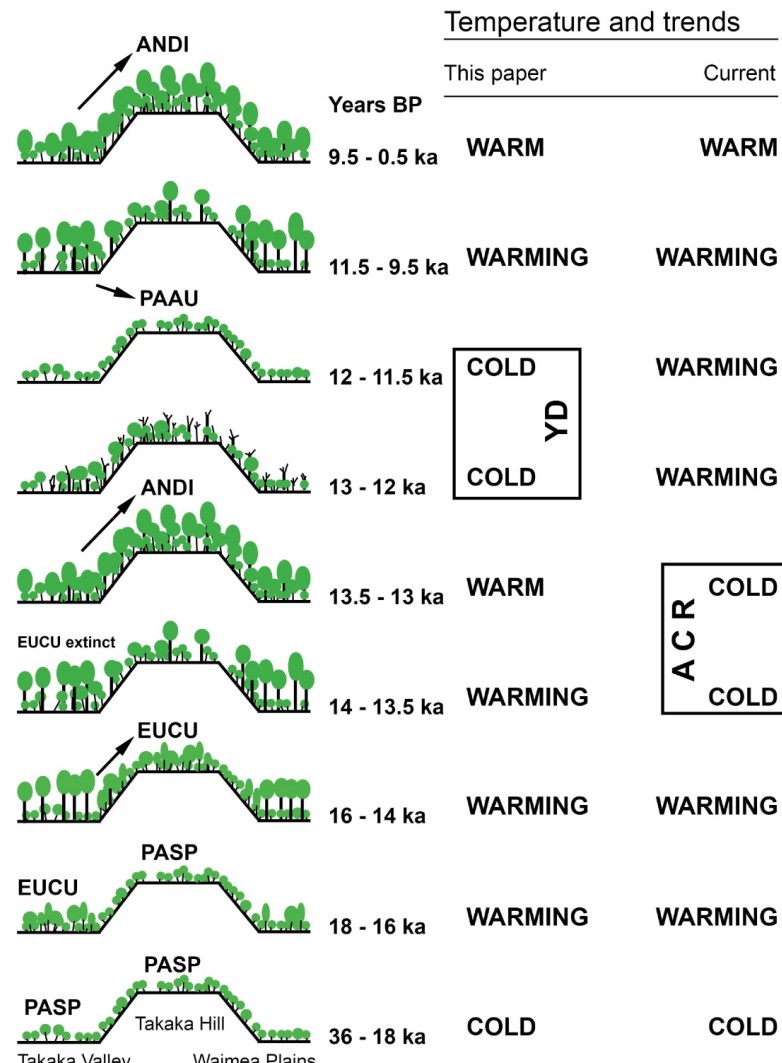


**Figure 12:** Comparison of present and proposed climatic contexts of changes in vegetation and moa populations
around Takaka 36 - 0.5 ka BP. Arrows indicate direction of spread of taxon during that period (downward indicates
movement from higher elevations to the south. YD, Younger Dryas; ACR, Antarctic Cold Reversal; other
abbreviations as in Fig. 5.
**4  Discussion**



Unlike indirect proxies such as oxygen isotope ratios and pollen samples from a few sites, moa were on-ground
real-time witnesses to the vegetation (and hence climate) at the time and place they lived and died. The congruence
of the high spatial and temporal resolution record of vegetation based on $^{14}$C ages of individual moa, each
characteristic of a particular vegetation (Fig. 12) with the local speleothem $\delta^{18}$O record suggest that both proxies
accurately reflect the timing of changes in the New Zealand glacial and post-glacial climate. However, the post-
glacial return to cold climate in both records is later than the ACR. Instead, the moa sequence accords with the $\delta^{18}$O
record in showing that interval to be one of warming climate at these southern latitudes.

The cold interval recorded in the $\delta^{18}$O and moa sequence on Takaka Hill, was coeval with the YD, when,

on the current model, the New Zealand climate should have been warming after the ACR. It is unlikely that there
are any significant issues with the moa radiocarbon ages which would affect their chronology because the ages from
different sites and measured by gas count and AMS provide coherent series.

In addition to the earlier evidence for a New Zealand YD presented by (Denton & Hendy, 1994) and (Ivy-

Ochs et al., 1999), more recently (Pauly et al., 2020) reported a return to cold climate in northern New Zealand
after the ACR, albeit of shorter duration. Stable isotope ratios in long-lived kauri (*Agathis australis*) trees reveal an
event that was brief relative to that in the north-western South Island. The difference may result from the trees'
being c. 5.5° (600 km) farther north, in an area surrounded by warm seas whose mild climate experiences, at least
today, few extremes of temperature (Chappell, 2013).

While the moa and speleothem $\delta^{18}$O chronologies both support a cold period synchronous with the

European YD, neither provides any information that would help to resolve the difference between their chronologies
and those derived from cosmogenic glacial moraine dating or Antarctic ice cores. However, as a similar disparity
between chronologies exists with cosmogenically dated advances (or stability at greater extent) of the North
Patagonia Ice Field at 47°S contemporary with the YD (Glasser et al., 2012). Although (Glasser et al., 2012) posit
higher precipitation as an alternative to cooler temperatures as the driver of the Patagonian ice advances, this
explanation would not account for the vegetation changes signaled by the sequence of moa observed on TH.
Increased rainfall there would not have driven a return to glacial shrublands, which would have required a period of
intense cold. Until the differences between present chronologies are resolved, the interhemispheric postglacial
climatic seesaw model provides only one interpretation of global post-glacial climate processes.

The Oruanui eruption stands at the onset of the Last Glacial Maximum in the moa species sequence and in

the local $\delta^{18}$O record. In addition to providing evidence for a Southern Hemisphere mid-latitude YD cold event, the





moa species sequence suggests that both the Oruanui super eruption of Taupo volcano (New Zealand) (Vandergoes
et al., 2013) affected the Southern Hemisphere climate. The climatic effects were sufficiently large to change the
vegetation pattern in the north-western South Island of New Zealand so that moa species changed their distributions.
The sudden appearance of *P. australis* at Honeycomb Hill immediately after the eruption means first that the species
was present in the area and second that there was a sudden increase in fossil deposition in the caves. That increase
may have resulted from an increase in local precipitation or from the removal of a pre-eruption vegetation cover and
its replacement by vegetation that was more susceptible to erosion.
(McConnell et al., 2017) suggested that the Mt Takahe eruption series in Antarctica may have initiated
post-glacial warming. In the moa species sequence, the eruption series marks a break in deposition of *P. australis* at
Honeycomb Hill. The break also coincides with a peak in the $\delta^{18}O$ record that began before the eruptions and which
declined rapidly after the eruptions. Instead, therefore, of the eruptions initiating the warming, they appear to have
stifled and reversed a warming trend, at least in central New Zealand. In initiating a cooling, these eruptions seem to
have had the same climatic effect of the much larger (VEI 8) – but much briefer – Oruanui eruption.
The patterns of moa species presence and absence west of New Zealand's South Island Main Divide appear
to provide a novel, valuable record of vegetation change over the past 30,000 years in a Southern Hemisphere
location athwart the mid-latitude westerlies. However, the patterns reported here were derived from published
radiocarbon ages measured for other purposes and not, therefore, aimed at providing complete chronologies for each
species in each site and each area. That coherent patterns are present that accord with the completely independent
climate proxy of speleothem $\delta^{18}O$ values suggests strongly the potential value of a concerted radiocarbon dating
programme of moa in developing a high resolution mid-latitude climate record for this most important, and
presently contentious, period.

**5 Conclusions**
• The patterns of presence and absence of habitat-specialist moa species in the north-western South Island of

New Zealand correspond to patterns in both an integrated multi-speleothem $\delta^{18}O$ record from the same area

and, significantly, in the Greenland GISP2 ice core proxies for the date of the European Younger Dryas.

• Neither the moa-based climate chronology nor that of the speleothem $\delta^{18}O$ values accords with that of the

Antarctic Cold Reversal.





• The present dichotomy in chronologies for post-glacial warming and cooling in the Southern Hemisphere
must be resolved before the interhemispheric postglacial climatic seesaw model can be accepted.
• More intensive, targeted radiocarbon dating of key moa species in the north-western South Island, New
Zealand will not only allow testing of the basis for the climate change chronology presented here, but could
provide a high precision chronology of climate in the region.


**6 Code availability**
No code was used.

**7 Data availability**
All data are included in the paper and the cited references.

**8 Author contributions**
RNH conceived the project, performed the analyses, drafted the figures, and prepared the manuscript.

**9 Competing interests**
The author declares that he has no conflict of interest.




**10 Acknowledgements**
Dr Richard Rowe (Research Associate, Australian National University) and Prof Don McFarlane (Claremont
Colleges, Los Angeles) provided helpful comments on the MS. There was no external funding for the study.







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
