# Peer review of "Palaeobiological evidence for Southern Hemisphere Younger Dryas and volcanogenic cold periods"

_Climate of the Past, 2021_

## Author Comment (AC1)

**Response to Reviewer 1**

I thank the reviewer for their rich and extended summary of the palynological and moraine literature on the New Zealand Late Quaternary. I will respond to those matters later but first, I will deal with the reviewer's brief references to the data and arguments presented in the paper itself.

The few sentences in the review devoted to the data reveal a fundamental misunderstanding of both moa biology and depositional regimes in New Zealand caves. Firstly, the reviewer contends that a few moa in a few caves does not constitute a climatic record because moa were large animals that freely wandered through the landscape. There is no evidence at all to support this statement: it is pure *ex cathedra* conjecture which could be seen as a throwaway line on which to reject the data. There is, of course, ample evidence that *populations* expanded and contracted their distributions over decades, centuries, and millennia, as would be expected in species responding to changing environments[1-9]. But there is no evidence at all that individual moa wandered across the landscape.

What little evidence that is available, from ancient genetics and trace elements from a species not referred to in the paper, suggests the contrary, i.e. that moa were sedentary. For example, closely related adult females of the South Island giant moa (*Dinornis robustus*) were deposited in the same site (Pyramid Valley) in the late Holocene[10]. An exploratory analysis of trace elements in the bones of giant moa from adjacent (< 6 km apart) sites in North Canterbury showed that all but one of the 11 had died in its natal area; the other one had moved only to the other site (Fig. 1). The issue of cave system "sampling" of moa faunas is covered below.

[Figure]

**Fig. 1** Principal component analysis of heavy metal composition of giant moa from Pyramid Valley (black) and Bell Hill Vineyard (white).

The reviewer also asserts that some moa were not confined to particular habitats. That is true, so far as it goes, but it does not include the taxa in the paper. Remains of giant moa have been found in situations where they would have occupied lowland rain forest, dry eastern forest, and high altitude shrubland and fellfield[5,6,11]. The upland moa, *Megalapteryx didinus* has also been found in areas of rain forest and high altitude shrubland and fellfield[1,11]. However, neither of these species is referred to in the paper. Of the moa actually referred to, the "crested" moa, *Pachyornis australis*, is well attested to have occupied glacial shrubland[12-14], and it "retreated" to high altitudes during the late Holocene[15,16].

For example[15], Abstract

> "We show that *Pachyornis* changed altitudinal, longitudinal and latitudinal ranges through the Late Quaternary in response to alterations in the distribution of suitable habitat….
> The results suggest that crested moa [i.e. *Pachyornis australis*] tracked habitat through time with little consequence to population size."

Also[1], page 224

> "At the higher altitude of Honeycomb Hill Cave, the Otiran fauna is dominated by M. didinus and P. australis, both of which were upland species (Worthy, 1990)"

And[6], page 217

> "The upland moa (*Megalapteryx didinus*) and the crested moa (*Pachyornis australis*) were found to be the predominant moas in a montane forest -subalpine scrubland habitat ==whereas the little bush moa *Anomalopteryx didiformis*== and large bush moa *D. novaezealandiae* [now the wet forest morph of both the North Island *D. novaezealandiae* and the South Island *D. robustus*) ==probably preferred wet dense lowland forests==. We suggested that *Pachyornis elephantopus* primarily preferred open dry habitats. These conclusions were further endorsed by (Worthy 1989a) and (Worthy 1989c). Also, I have (Worthy 1987) presented biological observations relating to the proposed ecology of *Pachyornis elephantopus* and *Euryapteryx geranoides* [now *E. curtus*] and concluded that their preferred habitat was dry lowland mosaics of shrubland, grasslands and forests."

In the Abstract[1]

> "The Holocene fauna [of the South Island West Coast, including *Anomalopteryx didiformis*] is assumed to have lived in vegetation similar to that found by the first Europeans, i.e. wet, dense, podocarp-hardwood forest, with swamp vegetation on riverbed flats."

As above, the little bush moa, *Anomalopteryx didiformis* has always and only been seen as a rain forest species[1-3,5,6], as cited above (highlighted). Lowland rain forest is still present in the central North Island "King Country", as it was throughout the Holocene, where Worthy[17] noted that:

> "*Anomalopteryx didiformis* dominated the moa assemblages in all sites, as it does in the King Country fauna as a whole (Millener 1981)."

As noted above in the extract from reference 6, *Euryapteryx curtus* was a dry forest and shrubland species that has never been found in association with either wet forest or at high altitudes[1,2,4-6,18].

Hence, there is no reason to suppose that the sequence of moa in the Takaka area represented anything other than a succession of replacement of populations.

Together, the extensive literature on moa distribution through space and time support my use of, primarily, *P. australis* and *A. didiformis,* as indicators of the presence of their characteristic habitats of alpine/glacial vegetation and lowland rain forest, respectively. Hence the reviewer's rejection of their use as environmental indicators has no basis.

**Samples from caves**

The second component of the reviewer's rejection of moa as environmental indicators is that of a perception of its being based on "a few moa in a few caves". This statement ignores the pattern and process of deposition in caves, and their role as natural "pit traps" collecting individuals from resident moa populations through time. The likelihood of the presence of "wanderers" amongst the residents therefore becomes amenable to basic statistical analysis.

More than 125 moa have been identified in large and small caves in the Takaka area. They were deposited in the caves, based on the oldest 14C-dated individual, over the past 30,000 years. That represents an average rate of c 4 per 1000 years, or one every 250 years. The probability that any of those birds being a wanderer from another area rather than being from a resident population whose members daily traversed the area around the caves is remote. For those extremely rarely trapped wanderers to be incorporated in exactly the pattern observed on Takaka Hill and in Takaka Valley is beyond remote.

The observed pattern, if generated by wandering birds, would require only wandering *A, didiformis* to be present and [14]C-dated before the Younger Dryas, only wandering *P. australis* to be trapped and [14]C-dated during the period of the Younger Dryas (among the local populations of whatever other resident taxa whose individuals were not trapped), and then only wandering *A. didiformis* to be trapped and dated thereafter. No *P. australis* wandered to Takaka Hill after the Younger Dryas from the populations which were definitely present in the mountains to the south throughout the Holocene[15,16].

These factors show that the sequence of habitat-specific moa in the Takaka area does indeed map a sequence of climate-driven vegetation types. The data therefore must be taken at face value. The discussion in the paper is an attempt to interpret the reality of the vegetation changes and their chronology in terms of data from other sources, which have their own issues, as below.

[Figure]

**Fig. 2** Probabilities of 1 or 2 wandering moa being incorporated in a Takaka Cave rather than a local population of 20 of another taxon

**The New Zealand pollen record**
The pollen record, comprehensively summarised by the reviewer, concentrates on taxa with high, often masting, pollen output, including podocarp trees, southern beeches, and *Chionochloa* grasses. No mention is made, in most instances, of the considerable number of insect- or bird-pollinated trees and shrubs. No attempt is made in New Zealand to weight the pollen record by pollen output, as is now standard practice in Europe and the United Kingdom. Examples of the changes in interpretation that can follow from weighting and factoring in habitat-related flowering and pollen production include[19,20] the replacement of the concept of a Holocene oak forest in England by that of a lime-dominated forest, and the recognition that hazel pollen in a deposits indicates the presence of hazel forest and not a hazel understorey as hazel does not flower under a canopy of other trees.

A very recent example from New Zealand of the potential mismatch between pollen and the distribution of vegetation, is the domination of the late Holocene vegetation of Central Otago by bird-pollinated *Sophora* trees, and not by anemophilous podocarp conifers[21].

Almost never mentioned are the results of studies of modern pollen distribution[22,23]. These suggest that pollen, particularly of some important taxa can accumulate far from their source(s). As can be seen from the cited references below, the reviewer is fully aware of these issues.

> From[22] , pages 263 and 271.
> "Sites above the treeline on the Main Divide showed anomalous high counts of exogenous Podocarpaceae pollen. This seems to be washed out of strong NW winds by orograpbic precipitation. Peat cores taken from 3 sites confirm that exogenous podocarp pollen influx has continued over at least the last 500 years.
>
> "The present study establishes some general principles about aerial pollen deposition across the Southern Alps. There is little evidence of podocarp pollen being deposited locally; instead it is carried by strong westerlies to the Main Divide where the high rainfall causes deposition of large amounts of it. Is this feature peculiar to upper valleys such as Otira? The phenomenon of transport of podocarp pollen upslope has been noted by previous workers (Moar, 1970; McGlone, 1982; Pocknall, 1982). Pocknall (1982) presumed it to be due to low pollen production of the local sub-alpine vegetation but the present results suggest that a different interpretation is warranted. Low podocarp pollen count values are apparent for all four Westland valley sites. Other observations in Westland (Pocknall, 1980) show podocarp pollen being deposited locally, although low frequencies of *Dacrydium cupressinum* have also been observed at some sites. These apparently anomalous results were considered by Pocknall (1978) to be due to the effects of vegetation structure and dispersal of large amounts of pollen of other taxa in the vegetation, but it appears that wind carriage of pollen to distant sites is a normal feature for this species. The dispersal pattern of *Nothofagus fusca* type pollen is such that it is not possible to distinguish between a few trees near the sample site and many trees at a great distance. *Nothofagus* pollen appears to be dispersed from east to west but only in trace frequencies. This is probably due to the predominant north-west wind flow. Poaceae pollen dominates in the grassland sites, with only low frequencies in forested areas. This has been considered to be due to only small amounts of Poaceae pollen being transported into and through forested sites (Pocknall, 1978). This study suggests that Poaceae pollen may not be well dispersed."

From[23] , page 215
"The accumulation *of Nothofagus fusca* type pollen increased with increasing distance from the forest, as in other wind-pollinated taxa (Davis et al., 1973). It was likely that within the forest, only the trees immediately around the site contributed pollen, whereas in open sites (e.g. at Lake Hawdon and Cave Stream) a larger number of trees contributed pollen. The accumulation data showed that there was a high accumulation of *Nothofagus* pollen in the grassland sites, representing only 12 to 15% of the total pollen. These results suggest that *Nothofagus* pollen was being dispersed into the grassland, but was masked by the local herbaceous pollen. This could explain Pocknall's (1982) observations of low percentage of *Nothofagus* pollen in the grassland sites, although beech forest was nearby."

*Moraines*

Early moraine studies in New Zealand also hypothesised a Younger Dryas[24], but these have been rejected more recently. However, moraine-based chronologies of glacial advances and retreats in the Southern Alps of New Zealand are perhaps not as straightforward to interpret as the reviewer implies. In particular, there is an ongoing discussion as to whether moraines can be distinguished from large rockfalls in the tectonically very active Southern Alps region[25-29].

In addition, the moraine chronologies can be based on assumptions that are somewhat circular. For example, from[30], page 17

"We made no corrections for snow cover or for erosion of boulder surfaces. In the central part of the Southern Alps, winter (June-July-August) snow cover is generally persistent only at altitudes above ~1500 m. Below that altitude a winter snowfall of 1 m is an exceptional event and generally melts away within a few weeks. Moreover, the sampled boulders protrude from the crests of moraine ridges and are likely to be swept clear of snow by the wind. Thus, at the elevation of our sample sites (1150 - 1450 m above sea level), significant shielding due to snow is unlikely, especially given the northerly (sunny) aspect of Reischek knob and Meins Knob."

The highlighted text assumes that present observations are directly relevant to the late glacial, which is by no means assured. The $^{10}$Be chronology itself, which is at variance with previous estimates of 10Be production[31] was based on a radiocarbon sequence from macrofossils[31] that may have been from (based on their figure 4) the same tree. None of the radiocarbon ages from those samples, and none cited as from sediments was assessed with respect to possible old carbon contamination from the soil[32].

With the timing of the environmental changes, it was natural to compare the sequence with those from other cave proxies from the same area (which lacks pollen records apart from a scanty one from Honeycomb Hill Cave over 50 km to the west of Takaka and on the other side of the northwest Nelson mountain ranges[14]. The most comprehensive of those records[33] related speleothem $\delta^{18}O$ values to temperature, at least in general terms, and a "significant negative excursion … spanned the Younger Dryas", as in the quote from the Abstract below:

> "Late-glacial warming commenced between 18.2 and 17.8 ka and accelerated after 16.7 ka, culminating in a positive excursion between 14.70 and 13.53 ka. This was followed by a significant negative excursion between 13.53 and 11.14 ka of up to 0.55x depth that overlapped the Antarctic Cold Reversal (ACR) and spanned the Younger Dryas (YD). Positive $\delta^{18}O$ excursions at 11.14 ka and 6.91–6.47 ka represent the warmest parts of the Holocene."

It was appropriate, therefore, to compare these results with the moa records from Takaka.

As the moa record and part at least of the speleothem record (above) both suggested a cool climate episode synchronous with the Younger Dryas, it was only appropriate then to compare the results with those showing an undoubted Younger Dryas. Hence the comparisons with the GISP 2 ice core records. Greenland was chosen because of the solid chronology for the cold period and because the land mass is, as is New Zealand, subject to westerly winds. Antarctica, by contrast, is isolated from the belt of strong westerly winds and has its own climate. There have been suggestions of a Younger Dryas event in South and Central America[34,35] on the other side of the Pacific, in South Africa[36], and in a Southern Ocean core[37].

**Summary**

The reviewer's bald rejection of the moa chronology for a Younger Dryas period in the northern South Island of New Zealand is based on a lack of understanding of the habitat requirements of the moa taxa involved, and of the processes of deposition in cave systems. Hence the data can be validly set against the pollen- and moraine-based chronologies and should not be just set aside because they run counter to current views.

**References**

Worthy, T.H. & Holdaway, R.N. 1993. Quaternary fossil faunas from caves in the Punakaiki area, West Coast, South Island, New Zealand. *Journal of the Royal Society of New Zealand* **23**, 147-254.
Worthy, T.H. & Holdaway, R.N. 1994. Quaternary fossil faunas from caves in Takaka valley and on Takaka Hill, northwest Nelson, South Island, New Zealand. *Journal of the Royal Society of New Zealand* **24**, 297-391.
Worthy, T.H. & Holdaway, R.N. 1995. Quaternary fossil faunas from caves on Mt Cookson, North Canterbury, South Island, New Zealand. *Journal of the Royal Society of New Zealand* **25**, 333-370, doi:10.1080/03014223.1995.9517494.
Worthy, T.H. & Holdaway, R.N. 1996. Quaternary fossil faunas, overlapping taphonomies, and palaeofaunal reconstruction in North Canterbury, South Island, New Zealand. *Journal of the Royal Society of New Zealand* **26**, 275-361.

Worthy, T.H. & Holdaway, R.N. *Lost world of the moa: prehistoric life in New Zealand.* (Indiana University Press and Canterbury University Press, 2002).

Worthy, T.H. 1990. An analysis of the distribution and relative abundance of moa species (Aves: Dinornithiformes). *New Zealand Journal of Zoology* **17**, 213-241.

Worthy, T.H. 1997. Quaternary fossil fauna of South Canterbury, South Island, New Zealand. *Journal of the Royal Society of New Zealand* **27**, 67-162.

Worthy, T.H. 1998. Quaternary fossil faunas of Otago, South Island, New Zealand. *Journal of the Royal Society of New Zealand* **28**, 421-521.

Worthy, T.H. 1998. The Quaternary fossil avifauna of Southland, South Island, New Zealand. *Journal of the Royal Society of New Zealand* **28**, 539-589.

Allentoft, M.E., Heller, R., Holdaway, R. & Bunce, M. 2015. Ancient DNA microsatellite analyses of the extinct New Zealand giant moa (*Dinornis robustus*) identify relatives within a single fossil site. *Heredity* **115**, 481-487.

Worthy, T. 1989. Moas of the subalpine zone. *Notornis* **36**, 191-196.

Worthy, T.H. 1989. Validation of *Pachyornis australis* Oliver (Aves; Dinornithiformes), a medium sized moa from the South Island, New Zealand. *New Zealand Journal of Geology and Geophysics* **32**, 255-266.

Worthy, T.H. *Fossils of Honeycomb Hill.* (Museum of New Zealand Te Papa Tongarewa, 1993).

Worthy, T.H. & Mildenhall, D. 1989. A late Otiran-Holocene paleoenvironment reconstruction based on cave excavations in northwest Nelson, New Zealand. *New Zealand Journal of Geology and Geophysics* **32**, 243-253.

Rawlence, N.J., Metcalf, J.L., Wood, J.R., Worthy, T.H., Austin, J.J. & Cooper, A. 2012. The effect of climate and environmental change on the megafaunal moa of New Zealand in the absence of humans. *Quaternary Science Reviews* **50**, 141-153, doi:10.1016/j.quascirev.2012.07.004.

Rawlence, N.J. & Cooper, A. 2013. Youngest reported radiocarbon age of a moa (Aves: Dinornithiformes) dated from a natural site in New Zealand. *Journal of the Royal Society of New Zealand* **43**, 100-107, doi:10.1080/03036758.2012.658817.

Worthy, T.H. & Swabey, S.E.J. 2002. Avifaunal changes revealed in Quaternary deposits near Waitomo Caves, North Island, New Zealand. *Journal of the Royal Society of New Zealand* **32**, 293-325.

Worthy, T.H. 1992. A re-examination of the species *Euryapteryx geranoides* (Owen) including comparisons with other emeiin moas (Aves: Dinornithiformes). *Journal of the Royal Society of New Zealand* **22**, 19-40.

Rackham, O. *Ancient woodland: its history, vegetation and uses in England.*  402 (Edward Arnold, 1980).

Rackham, O. *Woodlands.*  (Harper Collins, 2006).

Pole, M. 2022. A vanished ecosystem: *Sophora microphylla* (Kōwhai) dominated forest recorded in mid-late Holocene rock shelters in Central Otago, New Zealand. , (1):a1. https://doi.org/10.26879/1169 palaeo-electronica.org/ content/2022/3503-vanished-ecosystem. *Palaeontologia Electronica* **25**, doi:https://doi.org/10.26879/1169.

Randall, P. 1990. A study of modern pollen deposition, Southern Alps, South Island, New Zealand. *Review of Palaeobotany and Palynology* **64**, 263-272.

Randall, P. 1991. A one-year pollen trapping study in Westland and Canterbury, South Island, New Zealand. *Journal of the Royal Society of New Zealand* **21**, 201-218.

Denton, G.H. & Hendy, C.t. 1994. Younger Dryas age advance of Franz Josef glacier in the southern Alps of New Zealand. *Science* **264**, 1434-1437.

Tovar, D.S., Shulmeister, J. & Davies, T. 2008. Evidence for a landslide origin of New Zealand's Waiho Loop moraine. *Nature Geoscience* **1**, 524-526.

Shulmeister, J., Davies, T.R., Evans, D.J., Hyatt, O.M. & Tovar, D.S. 2009. Catastrophic landslides, glacier behaviour and moraine formation–A view from an active plate margin. *Quaternary Science Reviews* **28**, 1085-1096.

Reznichenko, N.V., Davies, T.R. & Alexander, D.J. 2011. Effects of rock avalanches on glacier behaviour and moraine formation. *Geomorphology* **132**, 327-338.

McColl, S. & Davies, T. 2011. Evidence for a rock-avalanche origin for 'The Hillocks''"moraine", Otago, New Zealand. *Geomorphology* **127**, 216-224.

Reznichenko, N.V., Davies, T.R. & Winkler, S. 2016. Revised palaeoclimatic significance of Mueller Glacier moraines, Southern Alps, New Zealand. *Earth Surface Processes and Landforms* **41**, 196-207.

Koffman, T.N., Schaefer, J.M., Putnam, A.E., Denton, G.H., Barrell, D.J., Rowan, A.V., Finkel, R.C., Rood, D.H., Schwartz, R. & Plummer, M.A. 2017. A beryllium-10 chronology of late-glacial moraines in the upper Rakaia valley, Southern Alps, New Zealand supports Southern-Hemisphere warming during the Younger Dryas. *Quaternary Science Reviews* **170**, 14-25.

Putnam, A., Schaefer, J., Barrell, D., Vandergoes, M., Denton, G., Kaplan, M., Finkel, R., Schwartz, R., Goehring, B. & Kelley, S. 2010. In situ cosmogenic 10Be production-rate calibration from the Southern Alps, New Zealand. *Quaternary Geochronology* **5**, 392-409.

Soter, S. 2011. Radiocarbon anomalies from old $CO_2$ in the soil and canopy air. *Radiocarbon* **53**, 55-69.

Williams, P.W., King, D., Zhao, J.-X. & Collerson, K. 2005. Late Pleistocene to Holocene composite speleothem 18O and 13C chronologies from South Island, New Zealand—did a global Younger Dryas really exist? *Earth and Planetary Science Letters* **230**, 301-317.

Heusser, C. & Rabassa, J. 1987. Cold climatic episode of Younger Dryas age in Tierra del Fuego. *Nature* **328**, 609-611.

Glasser, N.F., Harrison, S., Schnabel, C., Fabel, D. & Jansson, K.N. 2012. Younger Dryas and early Holocene age glacier advances in Patagonia. *Quaternary Science Reviews* **58**, 7-17.

Abell, P.I. & Plug, I. 2000. The Pleistocene/Holocene transition in South Africa: evidence for the Younger Dryas event. *Global and Planetary Change* **26**, 173-179.

Morigi, C., Capotondi, L., Giglio, F., Langone, L., Brilli, M., Turi, B. & Ravaioli, M. 2003. A possible record of the Younger Dryas event in deep-sea sediments of the Southern Ocean (Pacific sector). *Palaeogeography, Palaeoclimatology, Palaeoecology* **198**, 265-278.

---

## Author Comment (AC2)

**Response to Reviewer 2**

I greatly appreciate the effort and thought that the reviewer has put into their assessment of my manuscript. Some of their comments and concerns are dealt with in my response to Reviewer 1, but I will expand on those responses below. My responses are set out in relation to the specific sections which they concern.

> "For instance, the author argues than moa fossil assemblages are a more direct climate proxy than pollen or isotopes records, yet this hard to justify considering that the moa's habitat specificity could have certainly changed in the face of rapid landscape changes, and that the chronologies presented in the manuscript represent a discontinuous record of individual, ephemeral paleoecological events."

Over 35 years of research into moa biology shape my view that there is no evidence for any such change in moa habitat choice with environment change. Species changed their distribution with that of their preferred habitat[1,2]. All the evidence is consistent with a change in the composition of the moa fauna (and the assemblage associated with each fauna) in response to changes in the vegetation[1,3-18].. There is a substantial and fast-growing literature on the effect of current climate change on species distributions. Even subtle changes in a local environment can result in changes in abundance and the extirpation of taxa with specific habitat requirements[19], from which I quote: "Most could be accounted for by individual species' responses to events occurring primarily in the local breeding area. The most important local factor affecting bird abundance was temporal change in forest vegetation structure, resulting from natural forest succession and local disturbances. Four species that declined markedly and in some cases disappeared completely from the study plot…" For "natural forest succession" read significant climate-driven change from glacial/alpine shrubland to rain forest and back again.

> To sustain such a big claim, a convincing explanation about why significant warming (cooling) occurred in NZ during the ACR (YD) at the same time than several other mid-latitude terrestrial records were documenting significant cooling (warming) is essential. Unfortunately, it is missing in the text.

It is missing because the purpose of the paper is to present empirical evidence for changes in the local vegetation – as evidenced by the change in the moa species – that indicate a climatic reversal in central New Zealand contemporary with the Younger Dryas. I do not offer an explanation because I do not have one, except perhaps that New Zealand is an isolated land mass astride the (present) boundary between the circumpolar westerlies and subtropical air masses. However, do not I feel that an explanation is a necessary component of the paper. The evidence stands on its own and cannot be dismissed just because there is at present no explanation: that is surely a topic for future research.

> The results presented in the manuscript challenge the current consensus about the climate events that characterised LT in the Southern Hemisphere. In this regard, the author fails to place the NZ moa chronology within a continental paleoclimatic context. There is solid evidence, from a good number of high resolution, well-dated climate records, to sustain an extension of the Antarctic deglaciation pattern into the Southern Hemisphere mid-latitudes; however, only a very small number of studies are mentioned or discussed in the main text.

I do not agree that a challenge to a current consensus should be dismissed because it is a challenge. I cannot but agree that there are high resolution climatic records, but whether they sustain an extension of the Antarctic deglaciation to the mid latitudes might still be seen as a work in progress? Certainly the climatic reversal I report for central New Zealand might suggest that. My paper presents new evidence that cannot be dismissed on the grounds of unsubstantiated generalisations that moa could and did change habitat, when all other evidence, from different avenues of research, show that they did not.

> As the evidence presented in this manuscript is not discussed in the light of these (and several others) studies, the author omits an explanation for why the Moa chronologies suggest a warming-cooling pattern during the ACR-YD intervals, whilst terrestrial records from other southern mid-latitudes regions indicates the opposite climate pattern.

Other studies from low to mid southern latitudes have suggested a Younger Dryas cooling, even in New Zealand[20-25]. I am simply reporting that the moa chronologies indicate a warming-cooling pattern during the ACR-YD intervals. For many empirical observations, explanations come later.

> In addition, no Antarctic ice core data has been included in the figures or discussed in the main text. It is rather surprising that the manuscript places so much attention to the Greenland ice core data without mentioning or discussing the detailed Antarctic ice core isotope or gas timeseries. It seems that the author is over-stressing the data that agrees with its interpretation of the moa sequences (i.e., NZ speleothems, Greenland ice cores), to the detriment of a great number of detailed and well-dated records from NZ, the mid-latitudes, and Antarctica.

As I note in my response to Reviewer 1, I chose to concentrate on the Greenland record, not because the data agree with the northwest Nelson data, but because, first, Greenland shares a geographical position in a predominantly westerly air flow whereas Antarctica has its "own" climate south of the westerlies, but mainly, second, it was the obvious comparator for the chronology observed in northwest Nelson. It is a comparison of chronologies, not a proposition of cause and effect. The changes in vegetation recorded by the moa have a chronology that accords with that of the Younger Dryas, and only a direct comparison could confirm or refute that.

> The author indicates that -unlike pollen, cosmogenic chronologies, or speleothems- the radiocarbon record of fossil moa remains provides an unbiased and precise indication of climate variability during the LGM and the LT. In my opinion this assumption may be flawed, as the link between the presence/absence of moa species and climate conditions is in fact quite indirect.

It is in fact quite direct, and known to be so[1,2]. The moa populations "sampled" by deposition in the Takaka caves were resident: deposition was so rare (1 per 400-500 years on average) that it is extremely unlikely that wandering birds would have been preserved in preference to those residing around the caves, and that a succession of wanderers would have been preserved rather than residents is

[Figure]

**Fig. 1** Binomial probabilities for one or two glacial vegetation *Pachyornis australis* being deposited in preference to individuals of a resident rain forest *Anomalopteryx* moa population on Takaka Hill.

vanishingly small. The chance that alternations of wanderers would be preserved and then randomly appear in a dated sequence is next to zero (Fig. 1).

The appearance of the rain forest moa during the period of the ACR and of the glacial/alpine specialist during the Younger Dryas are both significant features in the sequence (Fig. 2).

Pollen can record both local and remote vegetation in New Zealand[26,27] as elsewhere, and there are no pollen records from northwest Nelson, apart from a limited record from a cave system on the western side of the northwest Nelson massif[3]. Speleothem and core isotopic data are must be related to local climate and vegetation by transfer functions. Moa were, as I maintain in the paper, direct and faithful witnesses to the vegetation around the caves at the time the birds were present, and hence of the climate at that time.

[Figure]

**Fig. 2** Poisson probabilities for rain forest *Anomalopteryx* and glacial vegetation *Pachyornis australis* on Takaka Hill. Note that occurrence of *Anomalopteryx* just after 15 ka BP and that of *P. australis* during YD period are both significant departures from random.

> For instance, *why moa remains are a better indicator of past vegetation than pollen assemblages?*

> Animals may change their diet specificity in response to climate alterations, and this could have certainly been the case for the moa species during the abrupt environmental changes while the world was thawing from the last glaciation. Hence, the changes in moa species during this time may be as an indirect climate proxy as pollen, isotopes, or other paleoclimate indicators.

It is not diet that is in question, but habitat preference for the two key species in the analysis. As noted, with copious referencing, in my response to Reviewer 1, *Pachyornis australis* is universally interpreted as requiring a cool climate vegetation. It retreated to high altitudes in the Holocene, to the extent that its survival was confirmed only recently by dating genetically identified individuals from caves in the mountains south of the Takaka area[2]. *Anomalopteryx didiformis* has for the past 30 years been identified as having been confined to rain forest[28].

While the author has made a great effort compiling a significant number of radiocarbon ages from moa fossil sites, some of the most critical inferences are based only on a small number of samples. For instance, the appearance of *Pachyornis australis* in the Takaka Hill site during the YD (indicative of cold/dry climates) is sustained just by two samples. Similarly, the responses to the Oruanui and Mt Takahe volcanic eruptions are inferred from a very limited number of radiocarbon dates.

Unfortunately, continual attempts at obtaining funding for a more complete series of radiocarbon ages have been unsuccessful. However, the dated individuals represent significant proportions of the available material from the cave systems. The statistics possible on these limited samples (e.g., Fig. 1, 2) support the conclusions on the timing of presence and absence of taxa in relation to the eruptions and the climatic events. In terms of radiocarbon ages, the moa in northwest Nelson are amongst the most intensively dated megafaunas globally.

I thank the reviewer for their detailed attention to the MS as listed under their **Minor changes**, and will certainly attend to them in a revised version.

However, I contest, as I have above, the comment on Lines 385-387. Moa chronologies are, I submit, not as indirect as pollen or isotopes. Moa were present at the site and were deposited from afar. Their presence does not reflect *changes in moa habitat preferences, driven by vegetation change.* This statement reflects a basic misunderstanding of moa biology (and the biology of many if not most birds): moa distribution changed in response to changes in the distribution of their required habitat (= vegetation). Vegetation, of course, directly reflects climate.

**References**

Rawlence, N.J., Metcalf, J.L., Wood, J.R., Worthy, T.H., Austin, J.J. & Cooper, A. 2012. The effect of climate and environmental change on the megafaunal moa of New Zealand in the absence of humans. *Quaternary Science Reviews* **50**, 141-153, doi:10.1016/j.quascirev.2012.07.004.

Rawlence, N.J. & Cooper, A. 2013. Youngest reported radiocarbon age of a moa (Aves: Dinornithiformes) dated from a natural site in New Zealand. *Journal of the Royal Society of New Zealand* **43**, 100-107, doi:10.1080/03036758.2012.658817.

Worthy, T.H. & Mildenhall, D. 1989. A late Otiran-Holocene paleoenvironment reconstruction based on cave excavations in northwest Nelson, New Zealand. *New Zealand Journal of Geology and Geophysics* **32**, 243-253.

Worthy, T.H. 1989. An analysis of moa bones (Aves: Dinornithiformes) from three lowland North Island swamp sites: Makirikiri, Riverlands and Takapau Road. *Journal of the Royal Society of New Zealand* **19**, 419-432.

Worthy, T. 1989. Moas of the subalpine zone. *Notornis* **36**, 191-196.

Worthy, T.H. 1990. An overview of the taxonomy, fossil history, biology and extinction of moas. *XX Congressus Internationalis Ornithologici* **1**, 555-562.

Worthy, T.H. 1990. An analysis of the distribution and relative abundance of moa species (Aves: Dinornithiformes). *New Zealand Journal of Zoology* **17**, 213-241.

Worthy, T.H. & Holdaway, R.N. 1993. Quaternary fossil faunas from caves in the Punakaiki area, West Coast, South Island, New Zealand. *Journal of the Royal Society of New Zealand* **23**, 147-254.

Worthy, T.H. 1994. Late Quaternary changes in the moa fauna (Aves: Dinornithiformes) on the West Coast of the South Island, New Zealand. *Records of the South Australian Museum* **27**, 125-134.

Worthy, T.H. & Holdaway, R.N. 1994. Quaternary fossil faunas from caves in Takaka valley and on Takaka Hill, northwest Nelson, South Island, New Zealand. *Journal of the Royal Society of New Zealand* **24**, 297-391.

Worthy, T.H. & Holdaway, R.N. 1995. Quaternary fossil faunas from caves on Mt Cookson, North Canterbury, South Island, New Zealand. *Journal of the Royal Society of New Zealand* **25**, 333-370, doi:10.1080/03014223.1995.9517494.

Worthy, T.H. & Holdaway, R.N. 1996. Quaternary fossil faunas, overlapping taphonomies, and palaeofaunal reconstruction in North Canterbury, South Island, New Zealand. *Journal of the Royal Society of New Zealand* **26**, 275-361.

Worthy, T.H. 1997. Quaternary fossil fauna of South Canterbury, South Island, New Zealand. *Journal of the Royal Society of New Zealand* **27**, 67-162.

Worthy, T.H. 1998. Quaternary fossil faunas of Otago, South Island, New Zealand. *Journal of the Royal Society of New Zealand* **28**, 421-521.

Worthy, T.H. 1998. The Quaternary fossil avifauna of Southland, South Island, New Zealand. *Journal of the Royal Society of New Zealand* **28**, 539-589.

Worthy, T.H. 2000. Two late-Glacial avifaunas from eastern North Island, New Zealand-Te Aute Swamp and Wheturau Quarry. *Journal of the Royal Society of New Zealand* **30**, 1-25.

Worthy, T.H. & Swabey, S.E.J. 2002. Avifaunal changes revealed in Quaternary deposits near Waitomo Caves, North Island, New Zealand. *Journal of the Royal Society of New Zealand* **32**, 293-325.

Attard, M.R., Wilson, L.A., Worthy, T.H., Scofield, P., Johnston, P., Parr, W.C. & Wroe, S. in *Proc. R. Soc. B.*  20152043 (The Royal Society).

Holmes, R.T. & Sherry, T.W. 2001. Thirty-Year Bird Population Trends in an Unfragmented Temperate Deciduous Forest: Importance of Habitat Change. *The Auk* **118**, 589-609, doi:10.1093/auk/118.3.589.

Glasser, N.F., Harrison, S., Schnabel, C., Fabel, D. & Jansson, K.N. 2012. Younger Dryas and early Holocene age glacier advances in Patagonia. *Quaternary Science Reviews* **58**, 7-17.

Heusser, C. & Rabassa, J. 1987. Cold climatic episode of Younger Dryas age in Tierra del Fuego. *Nature* **328**, 609-611.

Morigi, C., Capotondi, L., Giglio, F., Langone, L., Brilli, M., Turi, B. & Ravaioli, M. 2003. A possible record of the Younger Dryas event in deep-sea sediments of the Southern Ocean (Pacific sector). *Palaeogeography, Palaeoclimatology, Palaeoecology* **198**, 265-278.

Andres, M.S., Bernasconi, S.M., McKenzie, J.A. & Röhl, U. 2003. Southern Ocean deglacial record supports global Younger Dryas. *Earth and Planetary Science Letters* **216**, 515-524.

Clapperton, C.M., Hall, M., Mothes, P., Hole, M.J., Still, J.W., Helmens, K.F., Kuhry, P. & Gemmell, A.M. 1997. A Younger Dryas icecap in the equatorial Andes. *Quaternary Research* **47**, 13-28.

Denton, G.H. & Hendy, C.t. 1994. Younger Dryas age advance of Franz Josef glacier in the southern Alps of New Zealand. *Science* **264**, 1434-1437.

Randall, P. 1990. A study of modern pollen deposition, Southern Alps, South Island, New Zealand. *Review of Palaeobotany and Palynology* **64**, 263-272.

Randall, P. 1991. A one-year pollen trapping study in Westland and Canterbury, South Island, New Zealand. *Journal of the Royal Society of New Zealand* **21**, 201-218.

Worthy, T.H. & Holdaway, R.N. *Lost world of the moa: prehistoric life in New Zealand.* (Indiana University Press and Canterbury University Press, 2002).